# A β-catenin-driven switch in TCF/LEF transcription factor binding to DNA target sites promotes commitment of mammalian nephron progenitor cells

Qiuyu Guo[1], Albert Kim[1], Bin Li[2], Andrew Ransick[1], Helena Bugacov[1], Xi Chen[1], Nils Lindström[1], Aaron Brown[3], Leif Oxburgh[2], Bing Ren[4], Andrew P McMahon[1]*

[1]Department of Stem Cell Biology and Regenerative Medicine, Eli and Edythe Broad-CIRM Center for Regenerative Medicine and Stem Cell Research, Keck School of Medicine of the University of Southern California, Los Angeles, United States; [2]The Rogosin Institute, New York, United States; [3]Center for Molecular Medicine, Maine Medical Center Research Institute, Scarborough, United States; [4]Ludwig Institute for Cancer Research, Department of Cellular and Molecular Medicine, Institute of Genomic Medicine, Moores Cancer Center, University of California San Diego, San Diego, United States

**Abstract** The canonical Wnt pathway transcriptional co-activator β-catenin regulates self-renewal and differentiation of mammalian nephron progenitor cells (NPCs). We modulated β-catenin levels in NPC cultures using the GSK3 inhibitor CHIR99021 (CHIR) to examine opposing developmental actions of β-catenin. Low CHIR-mediated maintenance and expansion of NPCs are independent of direct engagement of TCF/LEF/β-catenin transcriptional complexes at low CHIR-dependent cell-cycle targets. In contrast, in high CHIR, TCF7/LEF1/β-catenin complexes replaced TCF7L1/TCF7L2 binding on enhancers of differentiation-promoting target genes. Chromosome confirmation studies showed pre-established promoter–enhancer connections to these target genes in NPCs. High CHIR-associated de novo looping was observed in positive transcriptional feedback regulation to the canonical Wnt pathway. Thus, β-catenin's direct transcriptional role is restricted to the induction of NPCs, where rising β-catenin levels switch inhibitory TCF7L1/TCF7L2 complexes to activating LEF1/TCF7 complexes at primed gene targets poised for rapid initiation of a nephrogenic program.

*For correspondence:
amcmahon@med.usc.edu

Competing interests: The authors declare that no competing interests exist.

## Introduction

The now classical model of canonical Wnt signaling invokes two transcriptional states (*Wiese et al., 2018*; *Steinhart and Angers, 2018*). In the absence of Wnt ligand, HMG box family Tcf transcription factors bind enhancers of Wnt target genes recruiting co-repressors (Tle, Ctbp, and others) to silence target gene expression. In the cytoplasm, the transcriptional co-activator β-catenin is phosphorylated by an axin/GSK3β-dependent β-catenin destruction complex, resulting in ubiquitin-mediated proteasomal degradation (*Schaefer and Peifer, 2019*). Upon Wnt ligand binding to Fzd receptor/Lrp co-receptors on the cell surface, the β-catenin destruction complex is sequestered to the activated receptor protein complex through axin interactions, removing β-catenin from GSK3β-directed, phosphorylation-mediated degradation (*Schaefer and Peifer, 2019*). As a result of increasing β-catenin levels, β-catenin is free to associate with TCF/LEF DNA binding partners, activating Wnt target gene transcription (*Mosimann et al., 2009*). While evidence suggests that all four mammalian Tcf family members are able to functionally interact with both Tle family co-repressors and β-catenin (*Brantjes et al., 2001*),

a variety of studies in a range of biological systems indicate that Tcf7l1 predominantly acts as a repressor, Tcf7l2 as a context-dependent activator or repressor, and Tcf7 and Lef1 as transcriptional activators, of Wnt target gene expression (*Lien and Fuchs, 2014*).

The adult (metanephric) mammalian kidney arises from distinct cell populations within the intermediate mesoderm (*McMahon, 2016*). All nephrons – the repeating functional unit of the kidney – arise from a small pool of a few hundred nephron progenitor cells (NPCs) established at the onset of kidney development (*Short et al., 2014*; *Kobayashi et al., 2008*). The subsequent balance in the maintenance, expansion, and commitment of NPCs is critical to ensuring a full complement of nephrons, approximately 14,000 in the mouse and 1 million in the human kidney (*Bertram et al., 2011*). A reduced nephron endowment has been associated with abnormal kidney function and disease susceptibility (*Luyckx and Brenner, 2010*; *McMahon, 2016*; *Bertram et al., 2011*). The maintenance and expansion of NPCs are supported by Fgf, Bmp, and Wnt signals produced by NPCs or adjacent mesenchymal interstitial progenitor and ureteric epithelial cell types (*McMahon, 2016*). Within this nephrogenic niche, ureteric epithelium-derived Wnt9b is thought to act on NPCs in a β-catenin-dependent transcriptional process to regulate NPC target gene expression and expansion of the nephron progenitor pool (*Karner et al., 2011*). The removal of *Wnt9b* from the ureteric epithelium and NPC-specific production of β-catenin also results in the failure of NPC differentiation (*Carroll et al., 2005*), whereas chemical inhibition of GSK3β (*Davies and Garrod, 1995*; *Kuure et al., 2007*), or genetic activation within NPCs of a β-catenin form insensitive to GSK phosphorylation-mediated proteasomal degradation, leads to Wnt9b-independent ectopic induction of differentiation-promoting gene targets (*Park et al., 2007*). Genetic and chemical modification of Wnt pathway components suggest different levels of β-catenin distinguish maintenance and commitment of NPCs (*Ramalingam et al., 2018*). Genomic analysis of β-catenin engagement at TCF/LEF recognition motifs within enhancers linked to genes driving NPC differentiation (*Park et al., 2012*), and subsequent transgenic studies demonstrating TCF/LEF-dependent activity of cis regulatory elements, provides strong evidence for a canonical Wnt/β-catenin/Tcf regulatory axis (*Mosimann et al., 2009*). Thus, canonical Wnt signaling directs opposing NPC programs: maintenance and expansion of uncommitted NPCs and their commitment to nephron formation.

In this study, we employed an in vitro model to investigate the genomic regulatory mechanisms underlying the diverse action of canonical Wnt signaling in NPC programs. In this system, maintenance and expansion of NPCs, or their commitment to a nephrogenic program, are controlled by varying levels of CHIR99021 (CHIR) (*Cohen and Goedert, 2004*) supplemented to a chemically defined nephron progenitor expansion medium (NPEM) (*Brown et al., 2015*). CHIR binding to Gsk3β inhibits Gsk3β-mediated phosphorylation and proteasomal degradation of β-catenin (*Yost et al., 1996*; *Aberle et al., 1997*). Analysis of chromatin interactions and TCF/LEF factor engagement at DNA targets supports a model where β-catenin levels act as a key regulatory switch to modify TCF/LEF complex engagement at DNA targets and commitment of NPCs to a nephron-forming program.

## Results

### Elevated CHIR levels mediate a rapid inductive response in mouse NPCs

A low level of CHIR (1.25 μM) is an essential component in NPEM medium supporting the expansion of NPCs while maintaining the nephron-forming competence (*Brown et al., 2015*). Within 3 days of elevating CHIR levels (3 μM), aggregate NPC cultures show a robust signature of nephron differentiation (*Brown et al., 2015*; *Ramalingam et al., 2018*). To develop this system further for detailed molecular characterization of CHIR/β-catenin-directed transcriptional events, we collected NPCs from E16.5 embryonic kidneys by magnetic-activated cell sorting (MACS; *Brown et al., 2015*). NPCs were cultured in NPEM supplemented with a maintenance level of CHIR (1.25 μM – low CHIR throughout) to promote self-renewal of NPCs. CHIR levels were then titrated to determine an effective concentration for a rapid activation of early target genes of NPC commitment, mirroring in vivo responses.

As expected, low CHIR conditions maintained Six2, a key determinant of the NPC state, but did not induce expression of Jag1 (*Figure 1A–C*, *Figure 1—figure supplement 1B*), a Notch pathway

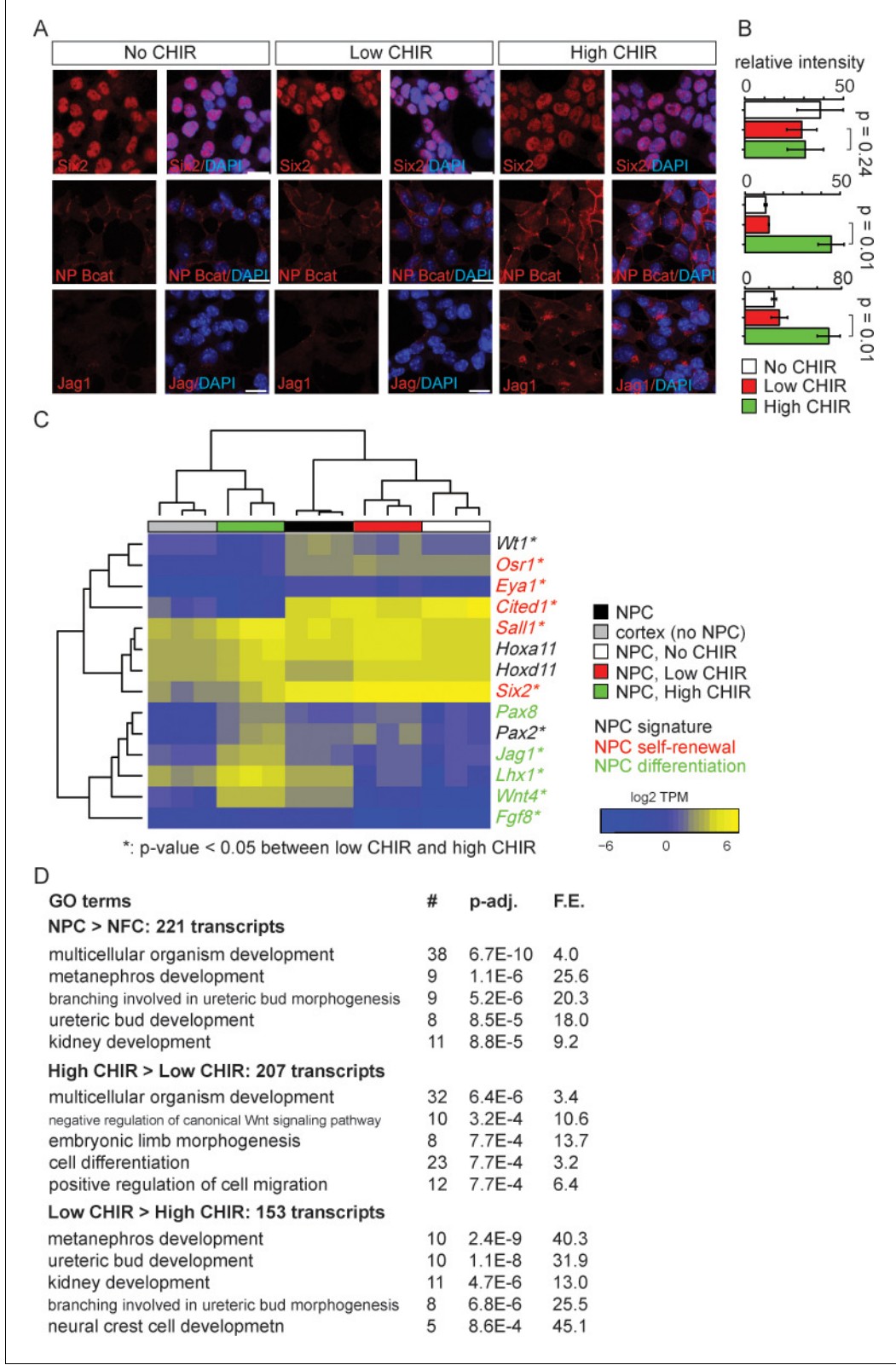

**Figure 1.** Nephron progenitor expansion medium (NPEM) supplemented with differential levels of CHIR99021 (CHIR) models nephron progenitor cell maintenance or differentiation in a plate. (**A**) Immunofluorescence (IF) staining showing expression level of Six2, non-phospho (NP) β-catenin and Jag1 in nephron progenitor cell (NPC) cultured in NPEM supplemented with various CHIR dosages. (**B**) Relative intensity

*Figure 1 continued on next page*

*Figure 1 continued*

of IF signals from individual cells in experiment associated with A. (**C**) Heatmap/hierarchical cluster of expression levels of NPC signature, self-renewal, and differentiation marker genes. (**D**) Top five enriched Gene Ontology (GO) terms of indicated differentially expressed gene lists, analyzed by DAVID. Link to high-definition figure: https://www.dropbox.com/s/76chyrs07m0toen/Fig%201.pdf?dl=0.

The online version of this article includes the following figure supplement(s) for figure 1:

**Figure supplement 1.** Supplementary RNA-Seq data analysis.

ligand activity at early stage of mouse and human NPC commitment (*Georgas et al., 2009*; *Lindström et al., 2018b*). A significant increase of cellular and nuclear β-catenin (*Figure 1A, B* and *Figure 3—figure supplement 1*) was observed in 5 µM CHIR ('high CHIR' throughout), along with a strong inductive response: Six2 protein level persisted, but there was a robust induction of Jag1 (*Figure 1A–C*), mirroring early inductive events in the pre-tubular aggregate and renal vesicle in vivo (*Lindström et al., 2018b*; *Mugford et al., 2009*; *Georgas et al., 2009*; *Xu et al., 2014a*). We adopted this induction condition throughout the study.

Next, we sought to systematically characterize gene expression profiles of NPCs in low and high CHIR conditions by mRNA-seq. Additionally, to explore the effect of low CHIR on NPCs, we generated data removing CHIR from the culture ('No CHIR' throughout) (*Figure 1—figure supplement 1A*). We also examined freshly isolated NPCs prior to culture. Low CHIR maintains expression of transcriptional regulators required for NPC specification and/or maintenance, including *Pax2*, *Wt1*, *Hoxa/d11*, and *Sall1* (*Figure 1C* and *Supplementary file 1*; *McMahon, 2016*). In contrast, high CHIR led to a downregulation of regulators and markers of self-renewing NPCs, including *Six2* (*Self et al., 2006*), *Cited1* (*Mugford et al., 2009*), *Osr1* (*Xu et al., 2014a*), and *Eya1* (*Xu et al., 2014b*), and a concomitant increase in expression of genes associated with induction of nephrogenesis, such as *Wnt4*, *Jag1*, *Lhx1*, and *Fgf8* (*Figure 1C*; *Park et al., 2007*). Trends in gene expression from mRNA-seq were confirmed by RT-qPCR analysis (*Figure 1—figure supplement 1B*).

To examine biological processes at play in different NPC culture conditions, we performed Gene Ontology (GO) enrichment analysis of differentially expressed genes (differential expression analysis described in 'Materials and methods') with DAVID (*Huang et al., 2009*). Comparing input NPCs freshly isolated from the kidney cortex with NPC-free cortical preparations (NFC), the strong enrichment for NPC-relevant GO terms (*Figure 1D*, top panel) was consistent with a strong enrichment of Six2+ NPCs (more than 90% of isolated cells were Six2+). When NPCs cultured in high versus low CHIR conditions were compared, a strong enrichment was observed in terms associated with Wnt signaling pathway, as expected, on CHIR-mediated NPC induction (*Figure 1D*, middle and bottom panels). Although primary NPCs, and their counterpart in low CHIR, showed similar expression patterns for NPC self-renewal and differentiation markers, transcriptome-wide comparison of all samples clustered primary NPCs into a distinct group from NPCs cultured in either low or high CHIR (*Figure 1—figure supplement 1C*). This is explained by a pronounced metabolic shift in culture where there is a strong enrichment in GO analysis in metabolic processes such as sterol biosynthesis (*Figure 1—figure supplement 1D*). In addition, freshly isolated NPCs showed a low-level inductive signature reflecting a co-contribution of small numbers of early induced NPCs (expression of *Fgf8*, *Wnt4*, *Lhx1*, *Heyl*, *Bmp4*, *Mafb*, and *Podxl*, *Figure 1C* and *Supplementary file 1*). Within 24 hr, low CHIR culture stabilized an undifferentiated NPCs signature with the downregulation of induction markers (*Figure 1C*, *Figure 1—figure supplement 1B*, and *Supplementary file 1*). Importantly, these data show that NPEM culture establishes a more rigorous model for distinguishing uninduced versus induced NPC responses than is possible with the intrinsic heterogeneity within primary isolates of NPC populations.

## CHIR-mediated induction modifies the epigenomic profile of NPCs

To investigate the chromatin landscape regulating NPCs, we integrated chromatin accessibility through ATAC-seq analysis (*Buenrostro et al., 2013*) with chromatin immunoprecipitation studies examining active (H3K27ac ChIP-seq) and repressive (H3K27me3 ChIP-seq) chromatin features, RNA Pol II recruitment (RNA Pol II Ser5P ChIP-Seq), and RNA-seq expression profiling (*Figure 1—figure supplement 1A*). Initially, we evaluated enhancers previously validated in transgenic studies

(*Park et al., 2012*) associated with *Six2* expression in uncommitted NPCs (Six2 distal enhancer; Six2DE) and *Wnt4* activation on NPC differentiation (Wnt4 distal enhancer; Wnt4DE). In low CHIR conditions, the Six2DE shows an open and active configuration: an ATAC-seq peak, flanked by H3K27ac peaks with Pol II engagement at the enhancer and within the gene body. In high CHIR, ATAC-seq, H3K27ac, and Pol II ChIP-Seq signals were reduced, correlating with the downregulation of gene expression (*Figure 2A*). As predicted, the Wnt4DE displayed an opposite trend in the shift from low to high CHIR: the ATAC and H3K27ac ChIP signals increased together with increased Pol II engagement in the gene body. Surprisingly, a marked enhancer-specific Pol II ChIP-seq signature was visible in low CHIR NPC maintenance conditions and reduced on initiation of active transcription in high CHIR (*Figure 2B*).

Having validated the data sets at target loci, we examined the data sets more systematically for broad features of epigenetic regulation. We focused on the differentially accessible regions (DARs) enriched in uncultured NPCs relative to NFC as they reflect a general NPC-specific signature. NPC-specific DARs were significantly enriched in transcription factor binding sites for Six, Pax, and Hox factors consistent with the critical roles of Six1/2, Pax2, and Hox11 paralogues in NPC programs (*Figure 2C*; *Naiman et al., 2017*; *Self et al., 2006*; *Wellik et al., 2002*). Functionally, GO-GREAT analysis (*McLean et al., 2010*) predicted that these regions were enriched near genes linked to kidney development (*Figure 2C*).

Hierarchical clustering of ATAC-Seq data was used to examine the relationship between CHIR dosage and the open chromatin landscape identifying DARs in NPCs cultured in low and high CHIR conditions (*Figure 2—figure supplement 1A*). Most of the DARs are distal to the transcriptional start site (TSS) of genes, indicative of enhancer elements (*Figure 2—figure supplement 1B*). The top two motifs identified in high CHIR-specific DARs were the Jun (AP-1) and TCF/LEF motifs (*Figure 2C*), supporting a β-catenin-driven increase in accessibility through engagement with TCF/LEF factors. Phosphorylated Jun has been detected in both NPCs and differentiates renal vesicles (*Muthukrishnan et al., 2015*). Further, Jun binds Tcf7l2 to cooperatively activate gene expression in Wnt-dependent intestinal tumors (*Nateri et al., 2005*). Elevated expression of *Jun, Junb,* and *Jund* in high CHIR (*Supplementary file 1*) suggests a potential interplay of Jun family members with β-catenin/Tcf-driven NPC differentiation. Statistical assessment (*Supplementary file 2*) determined that the Wnt4DE was significantly more accessible in high CHIR versus low CHIR conditions (peak ID 16567, adjusted p-value=0.02). The Six2DE showed a trend of greater accessibility in low CHIR versus high CHIR condition (peak ID 10164, adjusted p-value=0.13). These observations are consistent with differences in ATAC-Seq data comparing FACS-purified E16 and P2 Six2$^{GFP}$+ cells, where P2 NPCs are thought to have enhanced differentiation capability (*Hilliard et al., 2019*).

## Differential expression and DNA binding of TCF/LEF family members in the regulation of NPC programs

TCF/LEF factors directly bind to DNA and mediate the transcriptional response elicited by Wnt/β-catenin. Studies in other developmental systems have generally documented repressive roles for Tcf7l1 and Tcf7l2, and activating roles for Tcf7 and Lef1 (*Lien and Fuchs, 2014*). To examine the role of TCF/LEF factors directly in NPC maintenance and differentiation, we characterized expression of each of the four members (*Tcf7l1, Tcf7l2, Tcf7,* and *Lef1*). Of the four genes, *Tcf7l1, Tcf7l2,* and *Tcf7* transcripts were expressed at low (2–10 TPM; *Tcf7l2* and *Tcf7*) or moderate (50 TPM; *Tcf7l1*) levels in low CHIR NPC maintenance conditions. CHIR-mediated induction of NPCs resulted in a significant downregulation of *Tcf7l1* expression, while expression of both *Tcf7* and *Lef1* was markedly upregulated (*Figure 3A* and *Figure 1—figure supplement 1A,B'*). The same general trend was observed examining the level of each protein in the nucleus of NPCs through quantitative immunofluorescence (*Figure 3B, C*) and western blot (*Figure 3—figure supplement 1*) analyses. Interestingly, there is a significant decrease in Tcf7l2 protein but not transcript level in high CHIR versus low CHIR condition, suggesting transcriptionally independent regulation of Tcf7l2 activity. To compare in vitro findings with NPCs in vivo, single-cell RNA-seq (scRNA-Seq) transcriptional profiles were examined in cells isolated from E16.5 kidney cortex. In agreement with in vitro data, *Tcf7l1* transcripts were enriched in self-renewing NPCs while *Lef1* levels were elevated in differentiated NPCs, though *Tcf7* and *Tcf7l2* expression levels were relatively low, with little variation between non- and early induced NPC states (*Figure 3—figure supplement 2C–F*).

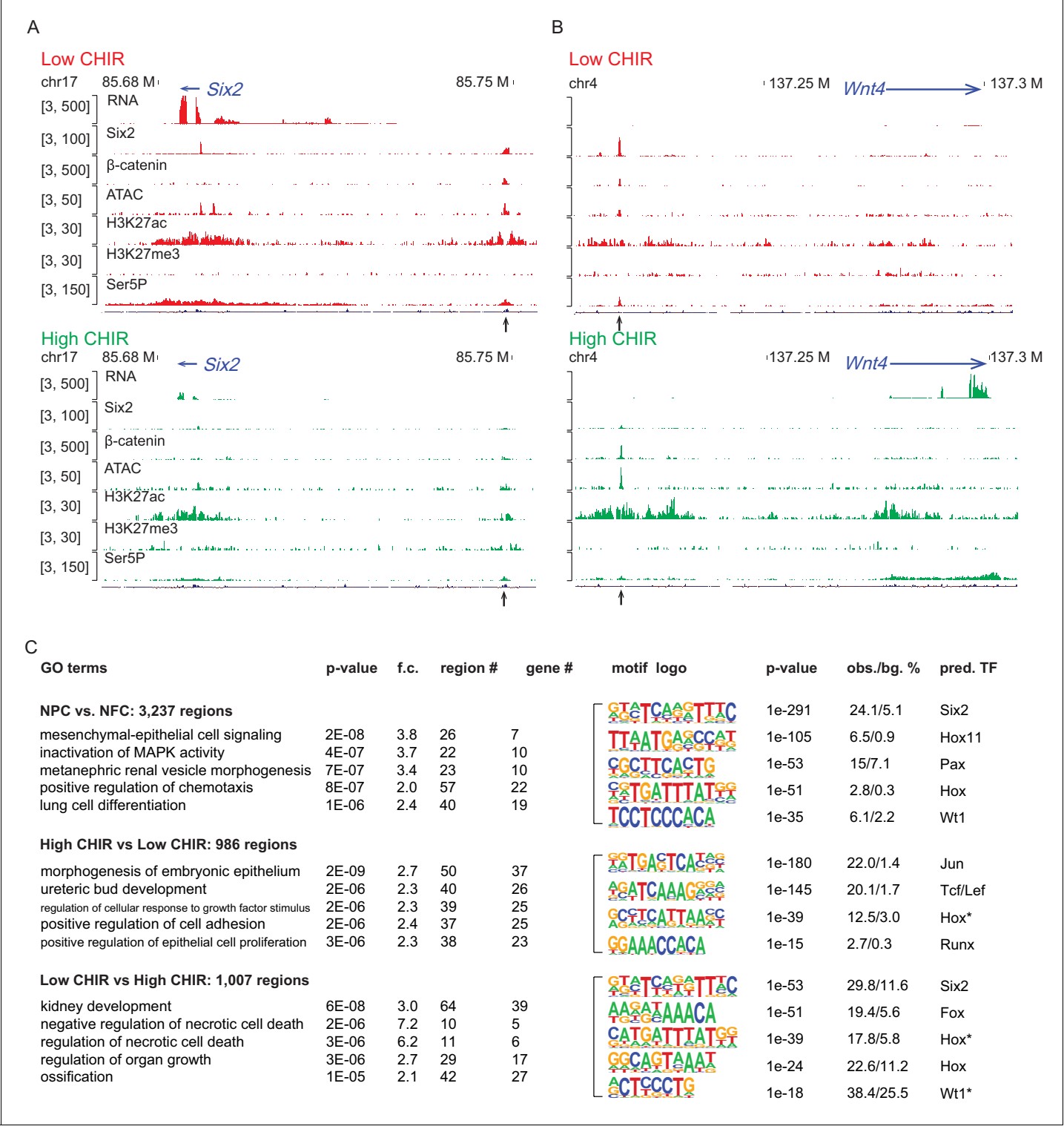

**Figure 2.** High dosage of CHIR99021 (CHIR) triggered change of nephron progenitor cell (NPC) epigenome. (**A, B**) Genome browser view of RNA-Seq, ATAC-Seq, as well as Six2, H3K27ac, H3K27me3, and Ser5P ChIP-Seq data near *Six2* (**A**) and *Wnt4* (**B**) in low CHIR (left) and high CHIR (right) conditions. Black arrow indicates Six2DE and Wnt4DE, respectively. (**C**) Display of differentially accessible regions (DARs) generated by the indicated comparisons. (Left) Heatmaps showing log2 normalized read counts of top 500 most significant DARs; (middle) top five most significant Gene Ontology (GO) terms associated with the DARs; (right) top five most enriched motifs discovered de novo in the DARs; * indicates less well-conserved motifs for the factor. NFC: NPC-free cortex.

*Figure 2 continued on next page*

*Figure 2 continued*

The online version of this article includes the following figure supplement(s) for figure 2:

**Figure supplement 1.** Supplementary ATAC-Seq data analysis.

To directly address TCF/LEF target interactions and β-catenin-mediated regulation of NPCs, we generated Lef1, Tcf7, Tcf7l1, Tcf7l2, and β-catenin ChIP-Seq data sets from freshly isolated, uncultured NPCs, and NPCs cultured in low and high CHIR (*Figure 1—figure supplement 1A*). Further, given the key role for Six2 in NPC maintenance and evidence supporting Six2 engagement in Tcf7l2 and β-catenin containing complexes (*Park et al., 2012*), we collected Six2 ChIP-seq data sets in the same conditions. Motif discovery of TCF/LEF/β-catenin ChIP-seq binding sites shows highest DNA

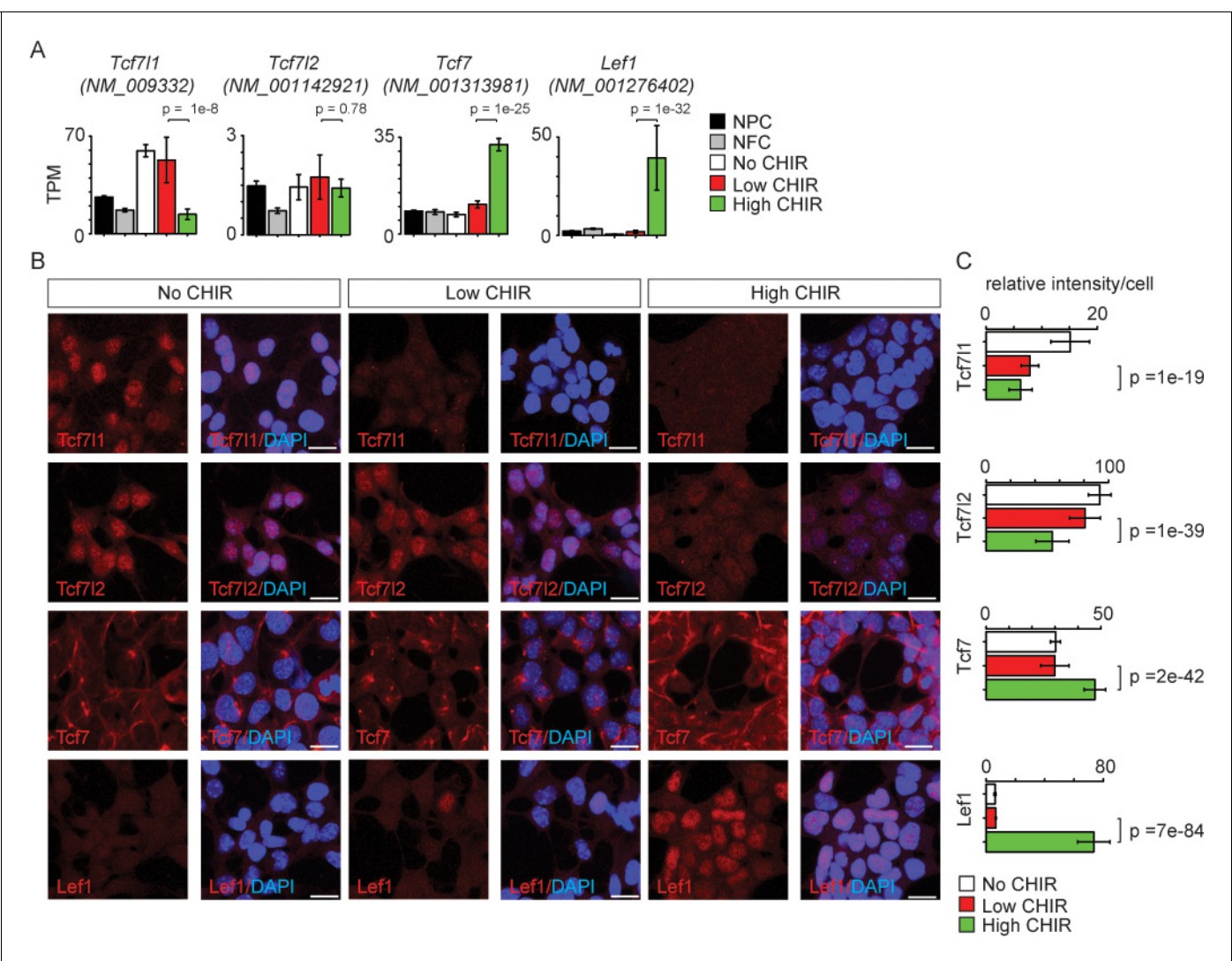

**Figure 3.** Differential expression of TCF/LEF family transcription factors in nephron progenitor cell (NPC) in response to distinct level of CHIR99021 (CHIR). (**A**) Bar plots showing RNA-Seq measured expression levels of TCF/LEF family factors in NPC cultured in nephron progenitor expansion medium (NPEM) culture supplemented with various concentrations of CHIR. (**B**) Immunofluorescence (IF) staining of TCF/LEF family factors in NPEM cultured with conditions indicated. (**C**) Relative intensity of IF signals from individual cells in experiment associated with B. Link to high-definition figure: https://www.dropbox.com/s/pvhu1ffhoujt39p/Fig%203.pdf?dl=0. NFC: NPC-free cortex.

The online version of this article includes the following figure supplement(s) for figure 3:

**Figure supplement 1.** Expression of TCF/LEF factors measured by immunoblots.
**Figure supplement 2.** Supplementary evidence for differential expression of TCF/LEF factors.

motif enrichment as the TCF/LEF binding element, indicating a high specificity of the data sets and supporting direct TCF/LEF/β-catenin target interactions (*Figure 4—figure supplement 1C*). In addition, a Hox motif is highly enriched in TCF/LEF binding sites in both low CHIR and high CHIR conditions consistent with Hox11 paralog regulation of the NPC state (*Wellik et al., 2002*; *Park et al., 2012*). A Runx motif is enriched in TCF/LEF binding sites in high CHIR. *Runx1* expression is upregulated in the same condition (*Supplementary file 1*), but the significance of possible Runx-dependent regulation remains to be determined.

Consistent with the different levels of each protein in low and high CHIR conditions, Tcf7l1 engagement at DNA targets was reduced on NPC induction, while Tcf7, Lef1, and β-catenin showed a marked increase in DNA bound sites (*Figure 4—figure supplement 1A*). In both low and high CHIR conditions, β-catenin association overlapped extensively with the binding of cognate TCF/LEF factors specifically enriched in each condition (*Figure 4—figure supplement 1B*). In either CHIR condition, motif recovery suggests direct engagement through TCF/LEF binding sites (*Figure 4—figure supplement 1C*). This is consistent with previous findings in murine intestinal studies that localization of β-catenin is primarily dependent on TCF/LEF factors (*Schuijers et al., 2014*).

Notably, hierarchical clustering indicates that the general feature of TCF/LEF factor binding in low CHIR and high CHIR was distinct and determined by the condition as different TCF/LEF factors targeted common genomic regions that differ between low CHIR and high CHIR conditions (*Figure 4—figure supplement 1D*). This observation was most evident examining Tcf7l2. *Tcf7l2* mRNA and protein levels did not vary greatly between low and high CHIR conditions but Tcf7l2 DNA interactions differed significantly (*Figure 4—figure supplement 1D*). Our previous studies identified (*Park et al., 2012*) and functionally validated (*O'Brien et al., 2018*) a Wnt4 distal enhancer (Wnt4DE, *Figure 2B*) driving *Wnt4* expression in response to NPC induction. This enhancer was shown to interact with Six2, Hoxd11, Osr1, and Wt1, critical determinants of the NPC state (*O'Brien et al., 2018*). Our data demonstrates that Tcf7l1 and Tcf7l2 bind to the Wnt4DE in low CHIR condition but were replaced by Tcf7 and Lef1 in high CHIR conditions (*Figure 4A*). This switch in TCF/LEF factor binding correlated with activation of *Wnt4* expression, consistent with a potential repressive role for Tcf7l1/Tcf7l2 interactions and activating role for Tcf7/Lef1 engagement (*Figure 4A*). Interestingly, β-catenin was engaged at the Wnt4DE in low CHIR condition where *Wnt4* was transcriptionally silent, though β-catenin binding at this enhancer was increased on high CHIR induction of NPCs (*Figure 4A*). In summary, though elevating β-catenin levels through high CHIR stabilization of GSK3β lead to some increase in binding of β-catenin at the *Wnt4* enhancer, a marked switch in the binding signature of TCF/LEF factors is a more striking correlation with subsequent activation of *Wnt4* transcription.

As Tcf7l1 and Lef1 binding best distinguished distinct NPCs' states, we performed a more extensive analysis of these factors. Comparing DNA regions bound by Tcf7l1 in low CHIR and those bound by Lef1 in high CHIR, we observed a significant overlap (p-value = 1e-569), though over half were unique to a data set (*Figure 4C*). We assigned the Tcf7l1/low CHIR-specific sites (see *Supplementary file 3*) as set 1 (also 'lost,' as no longer bound by TCF/LEF in high CHIR condition), overlapping sites as set 2 (also 'switch,' as TCF/LEF factors interchange at these sites between low and high CHIR conditions), and Lef1/high CHIR-specific sites as set 3 (also 'de novo,' as TCF/LEF binding sites arose on elevated CHIR levels). Interestingly, examining motif recovery across sets 1–3 indicated that only sets 2 and 3 showed a strong prediction for direct TCF/LEF binding (*Figure 4D*). Thus, Tcf7l1 likely associates with a large number of DNA regions without direct DNA binding at TCF/LEF response elements. The majority (73%) of Tcf7l1 sites bound in low CHIR with a predicted TCF/LEF motif overlapped with those bound by Lef1 in high CHIR (*Figure 4—figure supplement 2A*). In contrast, only 29% of Tcfl71 binding sites without a TCF/LEF motif prediction overlapped with Lef1 bound regions (*Figure 4—figure supplement 2A*). Thus, switching from a Tcf7l1 to a Lef1-centered DNA interaction at TCF/LEF response elements was a general feature of NPC induction response. Comparing with set 3 (de novo), set 2 (switch) sites displayed stronger binding of Tcf7l2, Tcf7, Lef1, and β-catenin, as well as higher level of markers of activated chromatin (ATAC-Seq and H3K27ac ChIP-Seq) in high CHIR condition (*Figure 4C*). Thus, the data suggests that TCF/LEF sites occupied by Tcf7l1 in low CHIR are poised for stronger binding (and therefore activation) by TCF/LEF activators in high CHIR condition. Interestingly, GO GREAT analysis of all three sets predicted expected kidney terms such as 'metanephric nephron morphogenesis' or 'renal vesicle morphogenesis' (*Figure 4E*), consistent with biologically relevant interactions, arguing against artifactual responses to culture or CHIR treatment. More than half of the potential target genes (assigned

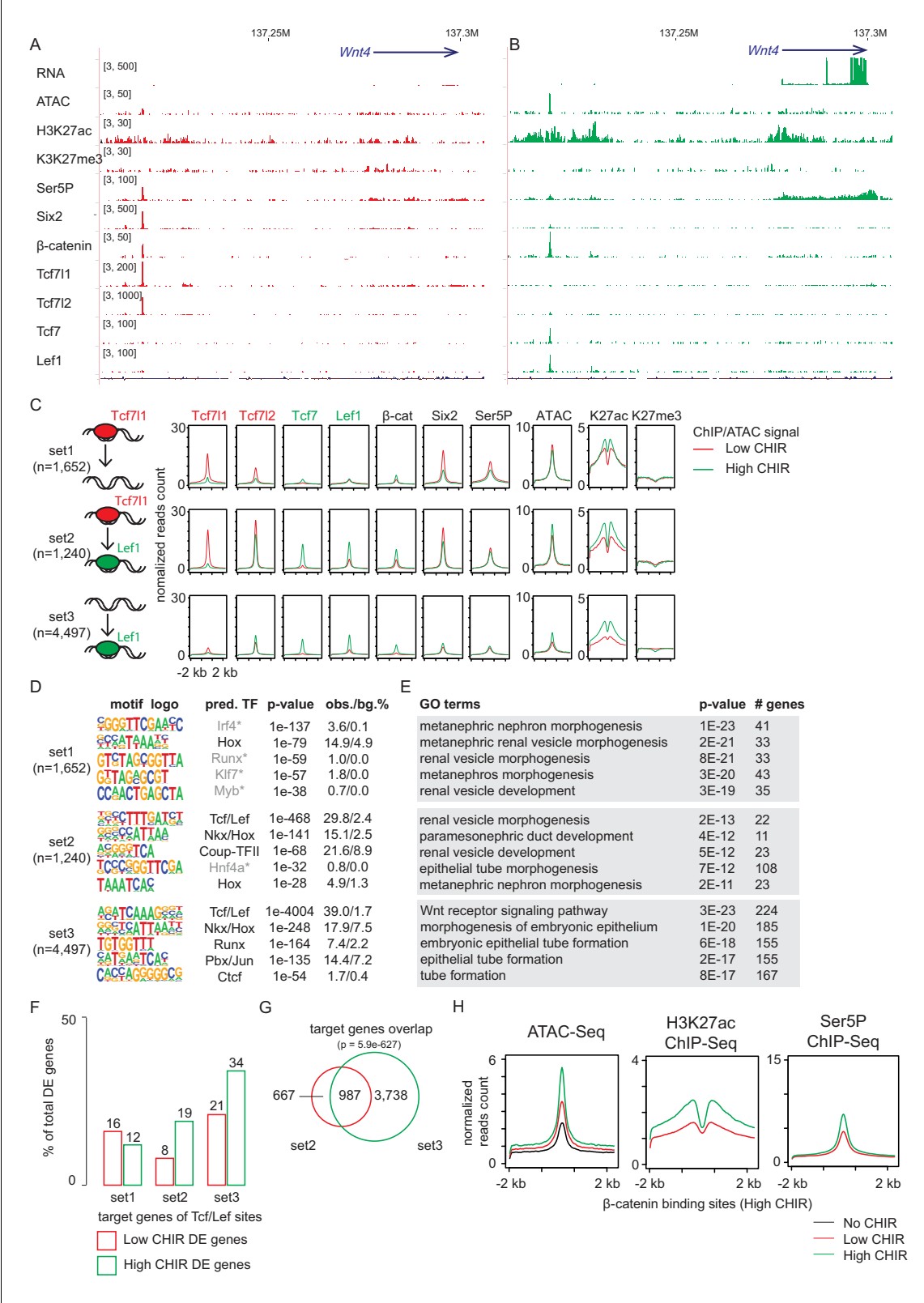

**Figure 4.** Increased CHIR99021 (CHIR) dosage induces a switch of TCF/LEF factors binding to the genome. (**A, B**) Genome browser view of *Wnt4* enhancer locus showing ChIP-Seq signal of TCF/LEF factors in low and high CHIR conditions. (**C**) Histograms showing binding intensity of TCF/LEF factors and chromatin markers on the three sets of TCF/LEF binding sites assigned by overlap of low CHIR Tcf7l1 and high CHIR Lef1 binding sites. (**D**) Result from de novo motif discovery of the three sets of TCF/LEF binding sites described in A. (**E**) Top Gene Ontology (GO) terms associated with the

*Figure 4 continued on next page*

*Figure 4 continued*

corresponding sets of TCF/LEF binding sites shown in B. (F) Percentage of TCF/LEF target genes belonging to different sets in differential expressed genes specific to low CHIR or high CHIR conditions as described in *Figure 1*. (G) Venn diagram showing overlap of set 2 and 3 target genes assigned by GREAT (*McLean et al., 2010*). (H) Histograms showing quantification of reads from the indicated data sets in ±2 kb of β-catenin binding sites in high CHIR condition.

The online version of this article includes the following figure supplement(s) for figure 4:

**Figure supplement 1.** Supplementary ChIP-Seq data analysis.
**Figure supplement 2.** Analysis of Tcf7l1 binding in low CHIR99021 (CHIR).
**Figure supplement 3.** β-Catenin binding sites near (A) Pla2g7 and (B) Tafa5, two β-catenin target genes reported in *Karner et al., 2011*.

bioinformatically by GREAT) of set 2 'switch' sites overlap with those of set 3 'de novo' sites (60%; *Figure 4G*, p=5.9e-627), although the latter are much broader. For this overlapping group, pre-engagement by Tcf7l1 may increase the likelihood of additional engagement in neighboring regions of DNA by Lef1 and potentially reinforce transcriptional input into a common gene target.

Examining chromatin features and RNA Pol II engagement, we observed stronger enrichment of active enhancer markers (H3K27ac and RNA Pol II) in low CHIR on set 1 sites predicting indirect Tcf7l1 engagement than set 2 sites, where Tcf7l1 was predicted to bind DNA directly through TCF/LEF motifs (*Figure 4—figure supplement 2E*). Thus, direct binding of TCF7L1/TCF7L2 on set 2 sites was associated with putative enhancers in a less active chromatin state in low CHIR condition. Consistent with this view, a direct analysis of the expression of target gene set of set 2 sites showed the highest expression in high CHIR conditions (*Figure 4F*), suggesting them tending to function in high CHIR. To examine whether elevated β-catenin correlates with chromatin activation, we examined change of active chromatin markers (ATAC-Seq and H3K27ac ChIP-Seq) and RNA Pol II loading on β-catenin bound sites. A dramatic elevation of all these signals in high CHIR condition (*Figure 4H*) is consistent with β-catenin engagement enhancing an open chromatin, active enhancer signature. Together, the data support the conclusion that high CHIR increased β-catenin association, correlating with switching of TCF/LEF factors at existing active chromatin, de novo opening of new chromatin sites, and an active transcription program initiating nephron formation.

## β-Catenin uses both pre-established and de novo enhancer–promoter loops to drive NPC differentiation program

To examine genome organization and chromatin interactions at a higher level in self-renewing and induced NPC, Hi-C analysis (*Rao et al., 2014*) was performed to characterize global chromatin–chromatin interactions. Further, as CTCF is known to mediate long-range chromatin interactions common across cell types, we integrated a published CTCF ChIP-seq data set generated from NPCs isolated directly from the developing kidney (*O'Brien et al., 2018*).

Analysis of two Hi-C data sets, using HiCCUPS within Juicer Tools (*Durand et al., 2016*), replicated 19,494 low CHIR and 20,729 high CHIR chromatin–chromatin interaction loops (*Figure 5—figure supplement 1A*). Almost half of all loops (40% low CHIR; 44% high CHIR), including those anchored on TSSs (35% low CHIR; 49% high CHIR), were unique to each culture condition (*Figure 5B* and *Supplementary file 4*). Given a focus on chromatin loops responding to β-catenin-mediated induction of NPCs, we examined loops anchored on β-catenin peaks identified in high CHIR conditions. Of the 5530 β-catenin-associated sites in high CHIR condition, 28% (1573) were located in loop anchors; 41% (647) of these connected to a TSS. Of the 647 peaks looped to TSSs, 57% (371) were connected by 'conserved' loops, shared between low CHIR and high CHIR conditions, the remainder appeared de novo under high CHIR conditions (*Figure 5A*).

To understand the biological consequences of these regulatory events, we identified promoters connected to β-catenin-bound enhancers (*Supplementary file 5*). Among those connected by conserved loops present in both CHIR conditions, 56 genes were highly expressed in the high CHIR condition (TPM >5; *Supplementary file 5*), including *Wnt4*, *Lhx1*, *Emx2*, *Bmp7*, and *Cxcr4*, which associate with NPC differentiation in vivo. Of these, only *Wnt4* enhancer elements have been rigorously defined through transgenic studies (*O'Brien et al., 2018*). In low CHIR conditions, Six2, Tcf7l1, Tcf7l2, and β-catenin bind to the *Wnt4* distal enhancer (Wnt4DE) while in high CHIR, Tcf7 and Lef1 replace Tcf7 and Tcf7l2, with a concomitant decrease in Six2 and increase in β-catenin association

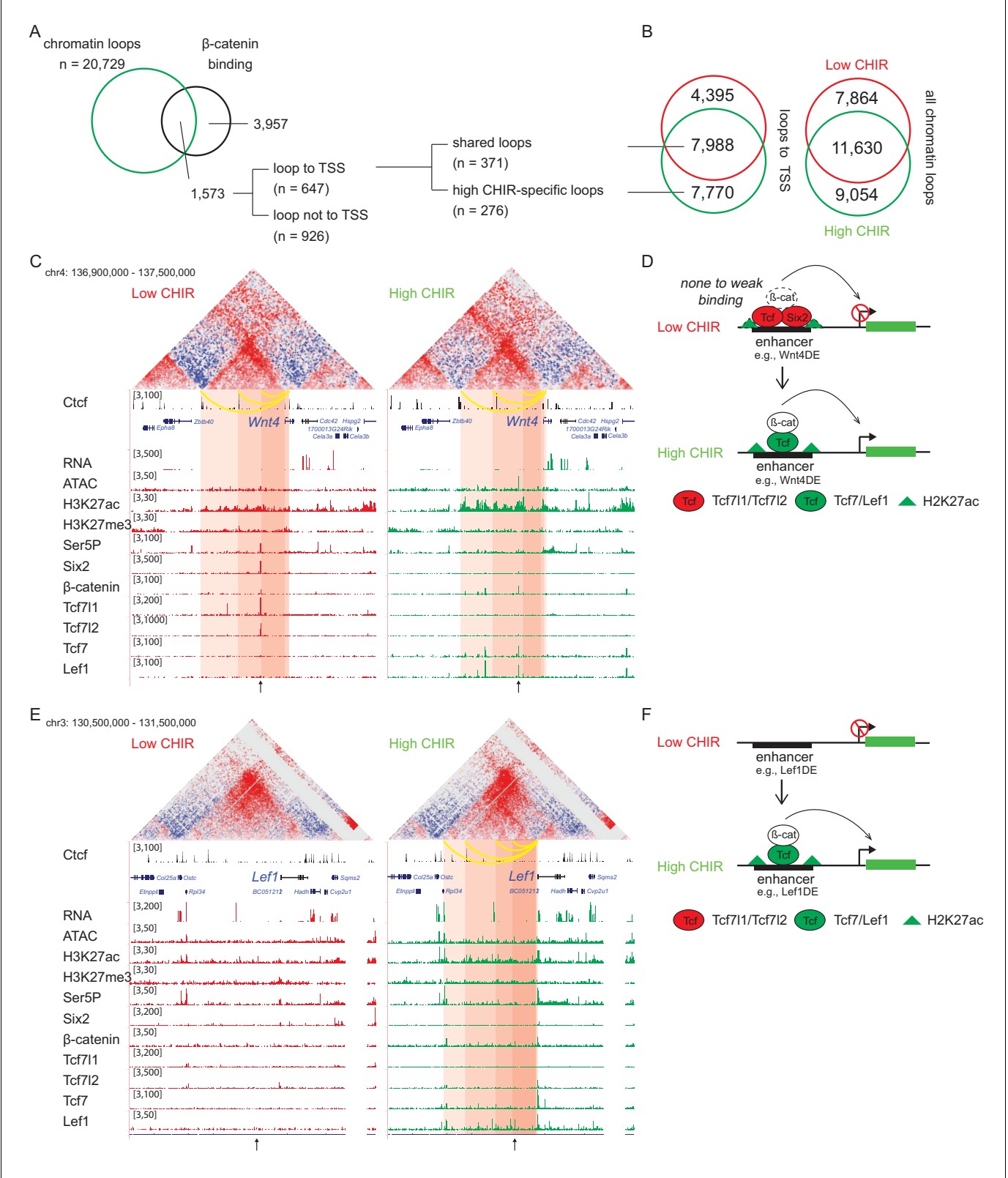

**Figure 5.** β-Catenin activates gene expression through both stable and de novo enhancer–promoter loops. (**A**) β-Catenin binding sites that overlap with an anchor of chromatin loop in high CHIR99021 (CHIR), the proportion that connects to a transcriptional start site (TSS) (gray in the pie chart) and segregation between two types of loops defined in B. (**B**) Overlap of chromatin loops between low CHIR and high CHIR conditions. Examples (**C, E**)

*Figure 5 continued on next page*

*Figure 5 continued*

and schematics (D, F) of β-catenin utilizing low/high CHIR-shared enhancer–promoter loops to activate *Wnt4* (C, D) or high CHIR-specific loops to activate *Lef1* (E, F). Black arrow at the bottom indicates the β-catenin binding sites involved in the loops.

The online version of this article includes the following figure supplement(s) for figure 5:

**Figure supplement 1.** Supplementary Hi-C data analysis.

(*Figure 5C*). Two additional loops further upstream anchor to Ctcf binding sites and loop to the *Wnt4* TSS (*Figure 5C*). Interestingly, all three loops are stable between low and high CHIR conditions consistent with a predetermined chromatin organization facilitating rapid activation of a nephron-forming inductive program, on switching the TCF/LEF transcriptional input (*Figure 5D*).

*Lef1* was among the gene sets defined by highly enriched expression in high CHIR conditions with de novo loop formation in high CHIR (*Figure 5E* and *Supplementary file 5*). The *Lef1* locus showed no loop connections to the *Lef1* TSS in low CHIR. Further, no TCF/LEF/β-catenin binding was detected, and ATAC-seq and H3K27 acetylation showed only background levels around the *Lef1* locus. In contrast, four interaction loops appeared in high CHIR conditions, one of which was associated with a strong Lef1/β-catenin interaction site, two mapped to CTCF bound regions (*Figure 5E*). In high CHIR, a string of LEF1/TCF7/β-catenin binding events accompanied enhanced accessibility (ATAC-seq) and the appearance of an active chromatin signature (H3K27ac) (*Figure 5E*). These data suggest that de novo loop formation may follow from de novo interaction of LEF1/TCF7/β-catenin binding complexes at the *Lef1* enhancer, in a feedback mechanism driving NPC commitment (*Figure 5F*).

## Discussion

Genetic analysis of mouse models points to a requirement for β-catenin in both the maintenance (*Karner et al., 2011*) and differentiation (*Park et al., 2007*) of NPCs. Genetically modulating Wnt inputs and chemically modifying Wnt-pathway activity suggest Wnt-modulation of β-catenin levels is critical to these alternative outcomes (*Ramalingam et al., 2018*). Previous studies directly examining β-catenin association in differentiating NPCs showed direct engagement of β-catenin at enhancers regulating expression of differentiation promoting genes such as *Wnt4* (*Park et al., 2012*). Data here confirmed these earlier findings and extended our understanding through a comprehensive analysis of Wnt-directed transcriptional engagement and epigenetic organization in a simple in vitro model of mammalian NPC programs. TCF/LEF factors are the transcription factors that ultimately mediate the transcriptional response to Wnt signaling. The direct analysis of all four TCF/LEF factors enables several key observations to be made, and conclusions drawn, from the analysis of NPC responses to CHIR modulation of β-catenin levels.

(1) In low CHIR, NPC maintenance and expansion conditions, the mitogenic activity mediated through CHIR stimulation of Wnt signaling appears to be independent of TCF/LEF binding to TCF/LEF motifs in target genes. (2) CHIR-mediated elevation of β-catenin is accompanied by reduced expression of mRNAs for transcriptional inhibitory TCF factors (*Tcf7l1* and *Tcf7l2*) and dramatic increase in expression of mRNAs for activating forms (*Tcf7* and *Lef1*), promoting an inductive program. (3) Direct binding of inhibitory Tcf7l1 and Tcf7l2 engagement at target motifs within putative cis-regulatory elements prefigures engagement of Tcf7 and Lef1 in a chromatin landscape primed for the transcriptional activation of the nephron-forming program. (4) High CHIR invokes a switch from inhibitory to activating TCF/LEF engagement at enhancers promoting nephron differentiation consistent with β-catenin controlling TCF/LEF target engagement. (5) In addition to enhancer–promoter loops pre-established in uncommitted NPCs associated with switching of inhibitory to activating TCF/LEF binding signatures on high CHIR induction, TCF/LEF/ β-catenin interactions at de novo sites may play additional roles in the inductive process, including positive feedback in the Lef1 program promoting nephrogenesis.

## Wnt signaling, β-catenin, and TCF/LEF factors in NPC maintenance and expansion

Though genetic evidence supports a direct role for β-catenin in regulating the maintenance and expansion of NPCs, and low CHIR activity is essential for normal NPC expansion in vitro, direct analysis of Tcf, Lef, and β-catenin engagement does not provide support for a direct transcriptional mechanism mediated through direct TCF/LEF DNA interactions with low CHIR-dependent target genes. We directly examined 16 genes identified in genetic screens in vivo to display *Wnt9b*-dependent expression (*Karner et al., 2011*). In total, 7 of the 16 showed elevated expression of at least one isoform in low CHIR versus no CHIR (*Figure 1—figure supplement 1G* and *Supplementary file 6*) consistent with a Wnt signaling input. Two of this group, *Pla2g7* and *Tafa5/Fam19a5*, were also reported to be ectopically activated by LiCl stimulation of the Wnt pathway and ectopic activation of β-catenin (*Karner et al., 2011*). Our data corroborated the upregulation of *Tafa5/Fam19a5* but not *Pla2g7* in high CHIR (*Figure 1—figure supplement 1G* and *Supplementary file 6*). However, ChIP-qPCR data arguing for β-catenin engagement around the TSS bindings sites of *Pla2g7* and *Fam19a5/Tafa5* could only be corroborated near one site in our data sets (*Figure 4—figure supplement 3*). Comparing NPC gene expression profiles in low CHIR versus no CHIR conditions revealed highly significant GO enrichment in cell cycle-related terms (*Figure 1—figure supplement 1E*), consistent with pro-proliferation roles of β-catenin in self-renewing NPCs. However, among the 85 genes associated with the term 'cell cycle,' only one gene showed significant β-catenin association within 500 kb of the TSS in low CHIR (*Figure 1—figure supplement 1F*), arguing against a scenario that directs transcriptional regulation of these genes through β-catenin engagement.

These findings raise the possibility that β-catenin acts through an alternative transcriptional mechanism. A prominent association of TCF7L1 is observed in low CHIR to DNA regions where the absence of a TCF/LEF motif may suggest indirect means of association, through protein–protein interactions. However, no strong consensus target emerges from examining motif enrichment in this subset of the TCF7L1 binding data (*Figure 4D*). Alternatively, β-catenin may play an essential, non-transcriptional role that links to control of cell proliferation. TCF-β-catenin nuclear complexes have been reported to oscillate with the cell cycle, suggesting potential nuclear roles independent of DNA association (*Ding et al., 2014*). Further, β-catenin is reported to play an essential, non-transcriptional role in self-renewal of mouse epiblast stem cells (*Kim et al., 2013*). Additionally, CHIR-mediated inhibition of GSK3 outside of β-catenin regulation might play a role. Interestingly, *Acebron et al., 2014* have reported that GSK3 inhibition by Wnt ligand administration leads to deubiquitylation and stabilization of target proteins. Importantly, our studies cannot exclude unknown complicating actions of β-catenin or GSK3-independent CHIR responses. Wnt-ligand mimetics and direct knockout of β-catenin activity offer a promising future approach to confirm CHIR-β-catenin-centered findings in the current study.

In mouse embryonic stem cell (ESC) culture, canonical Wnt signaling, induced by CHIR, supports long-term self-renewal of ESCs (*Ying et al., 2008*). Tcf7l1 has been shown to repress expression of genes involved in stem cell maintenance while Tcf7 and Lef1 activate targets (*Yi et al., 2011*; *Wray et al., 2011*) consistent with a classic canonical Wnt transcriptional activation program of stem cell renewal. However, other studies directly analyzing β-catenin interactions at the chromatin level suggest an indirect process where β-catenin may block the negative interplay of Tcf7l1 binding at the Sox motif of Sox-Oct bound stem cell promoting enhancers (*Zhang et al., 2013*). In this ESC system, β-catenin target sites containing TCF/LEF motif correlate strongly with differentiation promoting targets, consistent with the normal role of β-catenin in vivo in regulating gastrulation (*Haegel et al., 1995*). Addition of a second small molecule (PD03) inhibiting MEK/ERK signaling is essential to block this differentiation promoting activity (*Zhang et al., 2013*). Further, depletion of both *Tcf7l1* and *Tcf7* is sufficient to maintain Wnt ligand-independent expansion of ESCs, consistent with the implication that β-catenin activation can abrogate the repressive effect of Tcf7l11 independent of an activator Tcf7 (*Yi et al., 2011*).

In hair follicle stem cells (HFSC), Wnt induces a transition of the stem cells from the quiescent to the proliferative state. Tcf7l1 and Tcf7l2 are preferentially expressed in HFSC, while Lef1 and Tcf7 are preferentially expressed in the differentiated HFSC, that is, hair germ cells (*Merrill et al., 2001*; *Lien et al., 2014*). TCF7L1/TCF7L2 repress genes involved in HFSC differentiation, which are

activated by Tcf7/Lef1, correlating with replacement of TCF7L1/TCF7L2 by Tcf7/Lef1 on relevant enhancers, a close parallel to activity in NPC programs described here (*Adam et al., 2018*).

## Elevating β-catenin leads to activation of Tcf/Lef-bound enhancers

In the Wnt-off state, Tcf factors are known to recruit Groucho family co-repressors (*Cavallo et al., 1998*) and histone deacetylase (*Billin et al., 2000*) to repress Wnt target gene expression. Upon Wnt ligand stimulation, Groucho is replaced by β-catenin for activation (*Brantjes et al., 2001*; *Daniels and Weis, 2005*). From evidence in vitro, β-catenin has been shown to be able to recruit various chromatin modulators, including histone acetyl transferase (*Hecht et al., 2000*), histone methyl transferase (*Sierra et al., 2006*), and chromatin remodeler (*Barker et al., 2001*). Furthermore, through interaction with Pygo and Bcl9 (*Kramps et al., 2002*; *Schwab et al., 2007*), as well as direct interaction (*Kim et al., 2006*), β-catenin can form a complex with the Mediator complex, which bridges the Tcf-bound enhancer to RNA Pol II complex at the target gene promoter (*Jeronimo and Robert, 2017*). In addition, Six2 binding at certain TCF/LEF targets in NPCs might also confer a repressive effect; Six2 interacts with histone deacetylation complexes (HDACs), and depletion of the HDACs in NPC elevates *Wnt4* and *Lef1* expression (*Liu et al., 2018*). Similarly, the repressive chromatin modifiers *Ezh1* and *Ezh2* also repress Wnt4 and Lef1 expression in NPC through maintaining H3K27me3 mark (*Liu et al., 2020*), although the repressive functions of such chromatin modifiers tend to have broader effects.

The potential for an extended interaction of Tcf7l1 and Tcf7l2 with co-repressors in suppressing the NPC commitment program has not been addressed in this study. However, given the observation that activation of target genes committing NPCs to a nephrogenic program correlates with a switch to Tcf7 and Lef1 engagement, it seems unlikely that removal of a co-repressor input would be sufficient for full activation in the absence of these strong activators. The observed shift in TCF/LEF factor engagement at DNA targets through elevating CHIR levels raises the question of how rising levels of β-catenin might regulate this transcriptional switch. Given a dual role for Tcf7l2 as both a transcriptional repressor and activator in canonical Wnt transcription (*Korinek et al., 1997*; *Chodaparambil et al., 2014*; *Lien and Fuchs, 2014*), β-catenin may switch Tcf7l2 to an activator state. Alternatively, low levels of Tcf7 present in low CHIR may be sufficient for β-catenin engagement and transcriptional activation. In this model, binding of a Tcf7-β-catenin complex would be favored over inhibitory TCF complexes. In either scenario, it is likely that the transcriptional upregulation of *Tcf7* and *Lef1* creates a feed-forward loop to amplify the transcriptional activation response. Distinguishing among these possibilities will require effective and sensitive strategies to specifically modify regulatory components in the experimental model system.

## Pre-establishment of enhancer–promoter loops prefigures a nephrogenic program

Classical embryological studies have identified two broad categories of inductive processes by which uncommitted stem or progenitor cells make subsequent cell fate choices (*Saxén and Sariola, 1987*). Instructive signaling leads to cells adopting distinct cell fates each determined by the signaling input. In this scenario, cells have multiple fate choices. In contrast, permissive signaling only leads to a single outcome. The observation that nephron anlagen only undergo a restricted nephrogenic response, and no other, to inductive signals, was taken as evidence that NPCs are in an inflexible regulatory state, predetermined for kidney formation. Studies in *Drosophila* have indicated that during development certain enhancer–promoter loops are stable (*Ghavi-Helm et al., 2014*), that is, the enhancer–promoter loop is established before the target gene is activated as a mechanism to prime developmentally potent cells to differentiate into predestined cell fates.

Analysis of Hi-C data shows that enhancer–promoter loops present in low CHIR condition are consistent with NPCs exhibiting a primed genomic state promoting nephron-forming programs. Approximately 56% of loops observed in high CHIR were observed in low CHIR. Of these, 70% connected to a TSS (*Figure 5B*). Example of genes where such 'conserved' loops connect TCF/LEF/β-catenin binding events to TSS includes a number of genes within the nephrogenic program including *Wnt4*, *Lhx1*, *Emx2*, *Bmp7*, and *Cxcr4*. These data are consistent with the concept that at least a part of Wnt/β-catenin-activated differentiated program in NPC is primed through enhancer-TSS loop establishment, most likely at the time of specification of the NPC lineage, though these remain to be

determined. These findings also raise an interesting possibility that Tcf7l1 and Tcf7l2 engagement at such enhancer–promoter regions may maintain the primed state during an extensive period of progenitor expansion in the course of kidney development. Interestingly, in low CHIR maintenance conditions, the Wnt4DE showed higher levels of PolII association than in high CHIR conditions where *Wnt4* is transcribed (*Figure 2A, B*), consistent with a stable enhancer/promoter/PolII association in the primed state. Indeed, Tcf7l1 and Tcf7l2 might serve as a platform for assembly of the transcriptional machinery to facilitate target gene activation with a sufficient level of β-catenin. Hi-C studies also identify new loop interactions consistent with de novo gene activation, notably in a predicted Lef1 feed-forward loop.

In summary, NPC culture provides a powerful model for deepening a mechanistic understanding of the regulatory processes balancing maintenance, expansion, and commitment of NPCs. This is a valuable model for stem cells as bulk isolation of stem progenitor cells is problematic for many stem/progenitor systems. Further, in vitro conditions enabling the controlled switching between stem/progenitor and differentiation programs have only been described for a few of these systems. Given a broad role for Wnt/β-catenin signaling in regulating stem and progenitor cell programs in metazoans, the findings here may have broader significance for Wnt-directed control of organogenesis. Further, Notch and PI3K activity can also drive early nephrogenic responses in vivo or in vitro (*Lindström et al., 2015*; *Boyle et al., 2011*). Recent evidence also argues that not all NPCs entering the differentiation program differentiate. A minor subset returns to an NPC state after activating Wnt4, suggesting a variability in the epigenetic state among differentiating NPCs (*Lawlor et al., 2019*). The NPC culture model will provide a rigorous analytical platform for future exploration of how distinct pathway activities and variable epigenetic organization determine the induction of mammalian nephrons.

## Materials and methods

### Key resources table

| Reagent type (species) or resource | Designation | Source or reference | Identifiers | Additional information |
|---|---|---|---|---|
| Antibody | Rabbit polyclonal anti-Six2 | ProteinTech | 11562-1-AP; RRID:AB_2189084 | ChIP (1:200) |
| Antibody | Mouse monoclonal IgG1 anti-Six2 | Abnova | H00010736-M01; RRID:AB_436993 | IF (1:1000) |
| Antibody | Rabbit polyclonal anti-β-catenin | ThermoFisher | 71–2700; RRID:AB_2533982 | ChIP (1:200) |
| Antibody | Rabbit monoclonal anti-non-phopho-β-catenin | Cell Signaling Technology | 8814; RRID:AB_11127203 | ChIP (1:200), IF (1:500) and western blot (1:1000) |
| Antibody | Mouse monoclonal anti-Tcf7l1 | Santa Cruz | sc-166411; RRID:AB_2302942 | IF (1:1000), western blot (1:1000) |
| Antibody | Rabbit polyclonal anti-Tcf7l1 | Thermo Scientific | PA5-40327; RRID:AB_2577173 | ChIP (1:200) |
| Antibody | Rabbit monoclonal anti-Tcf7l2 | Cell Signaling Technology | 2569; RRID:AB_2199816 | ChIP (1:200), IF (1:500), western blot (1:1000) |
| Antibody | Rabbit monoclonal anti-Tcf7 | Cell Signaling Technology | 2203; RRID:AB_2199302 | ChIP (1:200), western blot (1:1000) |
| Antibody | Rat monoclonal IgG2b anti-Tcf7 | R and D Systems | MAB8224 | IF (1:100) |
| Antibody | Rabbit monoclonal anti-Lef1 | Cell Signaling Technology | 2230; RRID:AB_823558 | ChIP (1:200), IF (1:500), western blot (1:1000) |
| Antibody | Rabbit monoclonal anti-histone H3 | Abcam | ab1791; RRID:AB_302613 | Western blot (1:1000) |
| Antibody | Rabbit monoclonal anti-histone H3K27ac | Abcam | ab4729; RRID:AB_2118291 | ChIP (1:200) |
| Antibody | Mouse monoclonal anti-histone H3K27me2me3 | Active Motif | 39536; RRID:AB_2793247 | ChIP (1:200) |

*Continued on next page*

*Continued*

| Reagent type (species) or resource | Designation | Source or reference | Identifiers | Additional information |
|---|---|---|---|---|
| Antibody | Mouse monoclonal anti-Ser5P-RNAPII | Millipore | 05–623; RRID:AB_309852 | ChIP (1:200) |
| Antibody | Goat polyclonal anti-Jag1 | R and D Systems | AF599; RRID:AB_2128257 | IF (1:50) |
| Antibody | HRP-conjugated goat anti-mouse IgG1 | Thermo Scientific | A10551 | Western blot (1:1000) |
| Antibody | HRP-conjugated goat anti-rabbit IgG | Cell Signaling Technology | 7074 | Western blot (1:1000) |
| Antibody | Alexa647 mouse anti-rat IgG2b | Abcam | ab172335 | IF (1:500) |
| Antibody | Alexa555 donkey anti-rabbit IgG | Abcam | ab150074 | IF (1:500) |
| Antibody | Alexa488 goat anti-mouse IgG1 | Thermo Scientific | A10551 | IF (1:500) |
| Commercial assay or kit | SuperScript IV VILO Master Mix with ezDNase Enzyme | Thermo Fisher Scientific | 11766050 | |
| Commercial assay or kit | Luna Universal qPCR Master Mix | New England Biolab | M3003 | |
| Commercial assay or kit | RNeasy micro kit | Qiagen | 74004 | |
| Commercial assay or kit | KAPA Stranded mRNA-Seq Kit | Kapa Biosystems | KK8420 | |
| Commercial assay or kit | SimpleChIP chromatin IP buffers | Cell Signaling Technology | 14231 | |
| Commercial assay or kit | Protein A/G agarose beads | Thermo Fisher Scientific | 20423 | |
| Commercial assay or kit | minElute reaction cleanup kit | Qiagen | 28204 | |
| Commercial assay or kit | SEA block | Thermo Fisher Scientific | 107452659 | |
| Commercial assay or kit | Thruplex DNA library prep kit | Clontech | R400523 | |
| Software, algorithm | Partek Flow platform | Partek (https://www.partek.com/partek-flow/) | | |
| Software, algorithm | DESeq2 | *Love et al., 2014* | RRID:SCR_015687 | |
| Software, algorithm | DAVID | *Huang et al., 2009* (http://david.abcc.ncifcrf.gov/) | RRID:SCR_001881 | |
| Software, algorithm | Homer | *Heinz et al., 2010* (http://homer.ucsd.edu/) | RRID:SCR_010881 | |
| Software, algorithm | QuEST | *Valouev et al., 2008* | | |
| Software, algorithm | GREAT | *McLean et al., 2010* | | |

## NPC isolation and culture

NPEM formulation and NPC isolation followed the published protocol (*Brown et al., 2015*). Briefly, kidneys were harvested from fetal mice at E16.5 and placed into cold PBS. As each pair of kidneys was expected to yield approximately half a million NPCs, we routinely used 20 kidneys aiming for approximately 10 million NPCs. After collection, kidneys were washed with HBSS (Thermo Fisher Scientific, 14175-095), then incubated in 2 mL HBSS solution containing 2.5 mg/mL collagenase A (Roche, 11 088 793 001) and 10 mg/mL pancreatin (Sigma, P1625) for 15 min at 37°C while rocking on a Nutator platform at 250 rpm. The enzymatic reaction was then terminated by the addition of 0.5 mL of fetal bovine serum. The resulting supernatant was passed through a 40 μM filter, and then washed with AutoMACS running buffer (Miltenyi, 130-091-221) before spinning down at 500 g for 5 min. The cell pellet, predominantly cell of the cortical nephrogenic zone, was resuspended in 76 μL of AutoMACS running buffer for 10 million cells. NPC enrichment results from the removal of other cell types in the cell suspension using a combination of PE-conjugated antibodies as follows:

Anti-CD105-PE (Miltenyi, 130-102-548), 9 μL

Anti-CD140-PE (Miltenyi, 130-102-502), 9 µL
Anti-Ter119-PE (Miltenyi, 130-102-893), 8 µL
Anti-CD326-PE (Miltenyi, 130-102-265), 8 µL

The cells and antibodies were incubated at 4℃ for at least 30 min without agitation, then washed three times with 1 mL AutoMACS running buffer resuspending cells at each step in 80 µL the same buffer. To remove unwanted cells, 20 µL of anti-PE beads were added to the cell suspension for 30 min at 4℃, cells were washed three times in 1 mL of running buffer, and finally cells resuspended in 0.5 mL of AutoMACS running buffer and sent through the AutoMACS program as described in the published protocol to remove non-NPC cell types enriching for NPCs (>85%; *Figure 1—figure supplement 1H* and see *Brown et al., 2015* for additional details).

Ninety-six-well NPC culture plates were treated with Matrigel (Corning, 354277) 1:25 in APEL medium and incubated at room temperature in a laminar flow cabinet for cell culture for at least 1 hr. NPC Around 50,000 cells were added to a well for each imaging, RNA-seq or ATAC-seq study. For ChIP-Seq and Hi-C experiments, we cultured 1,500,000 cells in 6-well plates. For all culture experiments data was collected 24 hr after cell seeding.

## Reverse transcription followed by qPCR (RT-qPCR)

Total RNA was reverse-transcribed with SuperScript IV VILO Master Mix with ezDNase Enzyme (cat #: 11766050). qPCR was performed with Luna Universal qPCR Master Mix Protocol (New England Biolab #M3003) on a Roche LightCycler 96 System. p-Values were obtained by performing t-test between replicates of samples indicated. Primers used in RT-qPCR are listed as follows:

Six2:
 F: CACCTCCACAAGAATGAAAGCG
 R: CTCCGCCTCGATGTAGTGC
Cited1:
 F: AACCTTGGAGTGAAGGATCGC
 R: GTAGGAGAGCCTATTGGAGATGT
Wnt4:
 F: AGACGTGCGAGAAACTCAAAG
 R: GGAACTGGTATTGGCACTCCT
Jag1:
 F: CCTCGGGTCAGTTTGAGCTG
 R: CCTTGAGGCACACTTTGAAGTA
Fgf8:
 F: CCGAGGAGGGATCTAAGGAAC
 R: CTTCCAAAAGTATCGGTCTCCAC
Lhx1:
 F: CCCATCCTGGACCGTTTCC
 R: CGCTTGGAGAGATGCCCTG
Pax8:
 F: ATGCCTCACAACTCGATCAGA
 R: ATGCGTTGACGTACAACTTCT
Tcf7l1:
 F: CCCGCTGACACCTCTCATC
 R: ACAGTGGGTAATACGGTGACAG
Tcf7l2:
 F: AACGAACACAGCGAATGTTTCC
 R: CACCTTGTATGTAGCGAACGC
Tcf7:
 F: AACTGGCCCGCAAGGAAAG
 R: CTCCGGGTAAGTACCGAATGC
Lef1:
 F: TGTTTATCCCATCACGGGTGG
 R: CATGGAAGTGTCGCCTGACAG

## Immunofluorescence staining

To perform immunofluorescence staining, cell cultures were fixed with 4% PFA in PBS for 10 min, then washed with PBS twice before blocking in 1.5% SEA block (Thermo Fisher Scientific, 107452659) in TBST (0.1% Tween-20 in TBS). After minimally 30 min at room temperature, switched to primary antibody (diluted in blocking reagent) incubation in 4° overnight. After washing three times with TBST, switched to secondary antibody (diluted in blocking reagent) incubation for minimally 45 min in room temperature, blocking light. This was followed by three washes with TBST, then the cells were kept in PBS for confocal imaging. For freshly isolated cells (*Figure 1—figure supplement 1H*), cells were processed with Cytospin (Thermo Scientific) followed by the procedure described above. Quantification was performed with automatic program created in ImageJ.

## Immunoblots

To separate, protein samples containing at least 1 million cells were boiled with β-mercaptol and ran in SDS-PAGE gels casted from 30% acrylamide/Bis solution 29:1 (Bio-rad, 1610156) using the Mini-PROTEAN system (Bio-rad). Afterwards, the gel was transferred in Mini Trans-Blot Cell (Bio-rad) system to PVDF membranes (Immobilon-P, EMD Millipore, IPVH08100). The membrane with protein was blocked with I-block (Thermo Fisher Scientific, T2015) in TBST (0.1% TritonX-100 in TBS) at room temperature for 45 min before switching to primary antibody (diluted in blocking reagent) incubation in 4° overnight. Subsequently, the membrane was washed three times and switched to secondary antibody incubation (diluted in blocking reagent) for 45 min at room temperature. This was followed by three washes with TBST before drying the membrane and adding HRP substrate (Pierce ECL Plus Western Blotting Substrate, Thermo, 32132). Finally, the membrane was used on Autoradiography Film (5 × 7, Blue Devil, Premium, 100 Sheets/Unit, Genesee Scientific/Amazon) to visualize location of protein.

## mRNA-Seq and data analysis

In total, 50,000–100,000 cells were collected for each RNA experiment. RNA was isolated with RNeasy micro kit (Qiagen, #74004). mRNA-Seq libraries were prepared with KAPA Stranded mRNA-Seq Kit (Kapa Biosystems, #KK8420). The libraries were subsequently sequenced with Illumina Next-Seq500 model with pair-end 75 bp setting.

mRNA-Seq reads were aligned with STAR (*Dobin et al., 2013*) to mm10 assembly and quantified with Partek E/M to generate a count table, and finally converted to TPM for representation. All the steps above were implemented in the Partek Flow web platform (St. Louis, MO) sponsored by USC Norris Medical Library.

To identify differentially expressed genes, count tables of the two groups of data being compared were processed through DESeq2 (*Love et al., 2014*) to obtain the negative binomial p-values, which evaluates the significance of difference by read counts. The differentially expressed genes were defined with the following threshold: TPM >5, fold change >3, negative binomial p-value < 0.05, unless otherwise specified.

GO enrichment analysis was performed with DAVID (*Huang et al., 2009*).

## ChIP-Seq

1. *Fixation*. Freshly isolated NPCs were fixed in 1 mL AutoMACS running buffers (for each 3–5 million cells). Cultured NPCs were fixed in NPEM medium before scraping. In both cases, cells were fixed with final 1% formaldehyde (Thermo Fisher Scientific, #28908) for 20 min at room temperature.
2. *Chromatin preparation*. 3–5 million cells were processed for each chromatin preparation. Chromatin preparation includes cell lysis and nuclei lysis, which were done with SimpleChIP Sonication Cell and Nuclear Lysis Buffers (Cell Signaling Technology #81804) following manufacturer's instruction.
3. *Chromatin fragmentation*. For chromatin fragmentation, lysed nuclei were sonicated with Branson Ultrasonics Sonifier S-450 using a double-step microtip. Each sample was resuspended in 1 mL nuclear lysis buffer in a 15 mL conical tube, embedded in water-filled ice. Sonication was performed at 20% amplitude for 4 min, with 3 s of interval after each 1 s of duty time.
4. *Immunoprecipitation*. 1 million-equivalent fragmented chromatin was used for each immunoprecipitation experiment. Immunoprecipitation was done with SimpleChIP Chromatin IP

Buffers (Cell Signaling Technology #14231) following the manufacturer's instruction with the following details. The amounts of antibody used were case-dependent. In general, 2 μg or 1:50 to 1:100 antibody was used for each precipitation. Chromatin with antibody were rotated overnight at 4°C before 1:40 protein A/G agarose beads (Thermo Fisher Scientific, #20423) were added and incubated for another 6 hr to overnight. After washing and elution, antibody-precipitated input DNA were purified with minElute reaction cleanup kit (Qiagen, #28204), reconstituting to 35 μL EB buffer.

5. *ChIP-qPCR*. qPCR was performed with Luna Universal qPCR Master Mix Protocol (New England Biolab #M3003) on a Roche LightCycler 96 System. For each reaction, 0.5 out of 35 μL ChIP or input DNA was used. The qPCR primers used are listed below:

> Six2-DE:
> F: ggcccgggatgatacatta
> R: cgggtttccaatcaccatag
> Wnt4-DE:
> F: GACCCATAAGGCAGCATCCA
> R: CTTGCTGGGCAGAGATGAA
> Non-ChIP:
> F: tctgtgtcccatgacgaaaa
> R: ggaagtcatgtttggctggt

6. *Sequencing*. ChIP-Seq libraries were prepared with Thruplex DNA library prep kit (Clontech, #R400523). The libraries were sequenced with Illumina NextSeq500 model using single-end 75 bp setting.

## ChIP-Seq data analysis

ChIP-Seq reads were aligned with bowtie2. The alignment files are filtered to remove duplicate reads with Picard (http://broadinstitute.github.io/picard/index.html). Peak calling was performed with MACS2 (*Feng et al., 2012*) with combined replicate data sets of the ChIP/condition being considered and using combined replicate input from the same condition as control. To obtain relatively strong peaks, the peaks were first filtered for q-value <1e-4. Afterwards, the counts of normalized reads were generated within ±250 bp windows of the filtered peaks. To obtain consistent peaks in both replicates, we filtered the ones with >10-fold enrichment in the ±250 bp window in both replicates for downstream analysis. For data shown in Figure 4, the peaks were further filtered for those with fold enrichment >20 in order to focus on strong peaks. Overlapping peaks were defined as those within 150 bp from each other's center.

For visualization, wiggle tracks were generated with QuEST (*Valouev et al., 2008*). The intensity of peaks is measured as fold enrichment, which is calculated by the number of reads within the ±250 bp window divided by the total mapped reads in the library, normalized to the size of genome. De novo motif discovery and motif scan was performed with Homer (*Heinz et al., 2010*). GO analysis was performed with GREAT (*McLean et al., 2010*). To determine the overlap of ChIP-Seq peaks, peak centers from the two compared data sets were overlapped, and centers beyond 150 bp from each other were considered as unique sites (*Figure 4—figure supplement 1A, B*).

## ATAC-Seq and data analysis

Each ATAC-Seq experiment was performed with 50,000 cells, following the published protocol (*Buenrostro et al., 2013*). ATAC-Seq libraries were sequenced with Illumina NextSeq500 model using single-end 75 bp setting.

ATAC-Seq reads were aligned with bowtie2. Peak calling was performed with MACS2 (*Feng et al., 2012*) without control data. Subsequently, reads from each replicate were counted with Homer (annotatePeak.pl) within ± 250 bp of peak center. To obtain reproducible peaks, only the ones passing threshold (fold enrichment >3) in all three replicates were retained for downstream analysis.

To identify DARs, we merged reproducible ATAC-Seq peaks in two conditions and counted the reads within ±500 bp windows. DESeq2 was applied to identify statistically significantly DARs. The thresholds for identifying DARs are specified in *Supplementary file 1*.

To perform genome-wide hierarchical clustering, peaks from all replicate data sets in comparison were merged (peaks < 150 bp from each other are combined into one peak taking the midpoint as the new coordinate). Subsequently, ATAC-Seq reads from all samples concerned were counted in

±250 bp bins centering on the merged peaks, generating a count table. Hierarchical clustering was generated based on fold enrichment calculated from the count table.

De novo Motif discovery was performed with Homer (*Heinz et al., 2010*). GO analysis was performed with GREAT (*McLean et al., 2010*).

## Hi-C data generation and analysis

We generated about 700 million raw reads for each sample. The reads were aligned by bwa (*Li and Durbin, 2009*), then duplicates were removed with Picard. The Hi-C files were created and loop calling was done with the Juicer Tools (*Durand et al., 2016*).

To identify loops that are consistently present in both replicates, we extracted loops whose coordinates of anchors are within 10 kb between replicates.

| Condition | Replicate 1 | Replicate 2 | Overlap |
|-----------|-------------|-------------|---------|
| NPC-1.25  | 35854       | 39693       | 12181   |
| NPC-5     | 36731       | 40595       | 12231   |

To find TSS of genes and peaks connected by loops, we look for peaks that are within 5 kb from center of one of the loop anchors and TSS that are within 15 kb from center of the other loop anchor.

## Single-cell RNA-Seq and data analysis

Cortical cells were dissociated from E16.5 kidneys as described in *Brown et al., 2015*. Single-cell RNA-Seq library was synthesized using the 10X Genomics Chromium platform, with v2 chemistry and reads mapped with Cell Ranger, as described in *Lindström et al., 2018a*. Unsupervised clustering of transcriptional profiles, feature plots, and dot plots was generated with Seurat v2.3 (*Satija et al., 2015*). To computationally isolate the nephron lineage, we subset clusters of cells for those highly expressing nephron lineage markers (Six2, Wnt4, Wt1) but not interstitial progenitor cells markers (Foxd1, Meis1, Fat4).

## Sample size estimation

To enable calculation of standard error and finding of statistically differentially expressed genes and DARs, we generated three samples per condition for RNA-Seq and ATAC-Seq; due to restriction on resource, we generated two samples per condition for ChIP-Seq and Hi-C experiments.

## Replicates

Each replicate experiment was performed with a 1–2-month interval. To us, biological replicates are the same assays performed over different litters of mice, and technical replicates are the same biological material (e.g., DNA and RNA) that were assayed multiple times. For sequencing data (stated above), we generated biological replicates only and no technical replicates. For qPCR, we did generate three technical replicates per biological replicate, with three biological replicates per condition in total. We did not encounter any outliers and exclude any data.

## Data depository

All of our sequencing data, except for the scRNA-Seq data, has been submitted to GEO (GSE131119).

## Additional information

### Funding

| Funder | Grant reference number | Author |
|--------|------------------------|--------|
| National Institute of Diabetes and Digestive and Kidney Diseases | R01 DK054364 Cell Interaction in Development of the Mammalian Kidney | Andrew P McMahon |

The funders had no role in study design, data collection and interpretation, or the decision to submit the work for publication.

## Author contributions

Qiuyu Guo, Conceptualization, Data curation, Formal analysis, Validation, Investigation, Visualization, Methodology, Writing - original draft, Writing - review and editing; Albert Kim, Conceptualization, Formal analysis, Validation, Investigation, Visualization, Methodology; Bin Li, Data curation, Formal analysis, Methodology; Andrew Ransick, Data curation, Formal analysis, Investigation, Methodology; Helena Bugacov, Writing - review and editing; Xi Chen, Nils Lindström, Bing Ren, Methodology, Writing - review and editing; Aaron Brown, Resources, Methodology; Leif Oxburgh, Resources, Methodology, Writing - review and editing; Andrew P McMahon, Conceptualization, Supervision, Funding acquisition, Writing - review and editing

## Author ORCIDs

Qiuyu Guo  https://orcid.org/0000-0001-8549-8335
Andrew P McMahon  https://orcid.org/0000-0002-3779-1729

## Ethics

Animal experimentation: This study was performed in strict accordance with the recommendations in the Guide for the Care and Use of Laboratory Animals of the National Institutes of Health. All of the animals were handled according to approved institutional animal care and use committee (IACUC) protocols (#11893) of the University of Southern California. The protocol was approved by the Committee on the Ethics of Animal Experiments of the University of Southern California. Every effort was made to minimize suffering.

## Decision letter and Author response

Decision letter https://doi.org/10.7554/eLife.64444.sa1
Author response https://doi.org/10.7554/eLife.64444.sa2

# Additional files

## Supplementary files

- Supplementary file 1. Differentially expressed genes (DEG) between conditions.
- Supplementary file 2. Differentially accessible regions (DAR) between conditions.
- Supplementary file 3. Tcf7l1 and Lef1 peak coordinates by category as in *Figure 4*.
- Supplementary file 4. Hi-C loops to transcriptional start site (TSS).
- Supplementary file 5. Hi-C loops that are anchored on β-catenin binding sites in high CHIR99021 (CHIR) condition.
- Supplementary file 6. Expression of Wnt target genes published in *Karner et al., 2011*.
- Transparent reporting form

## Data availability

All RNA-Seq, ATAC-Seq, ChIP-Seq and HiC data sets are accessible through GEO (GSE131119).

The following dataset was generated:

| Author(s) | Year | Dataset title | Dataset URL | Database and Identifier |
|---|---|---|---|---|
| Guo Q, Kim AD, McMahon AP | 2021 | Genome-wide map of open chromatin, trancription factor occupancy, chromatin marks and gene expression profiles in mouse nephron progenitor cells | https://www.ncbi.nlm.nih.gov/geo/query/acc.cgi?acc=GSE131119 | NCBI Gene Expression Omnibus, GSE131119 |

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
