## [Decision Letter]

**Acceptance summary:**

We are excited to publish your findings and believe your study will significantly enhance our understanding of the dosage dependent role for Wnt signaling in stem cell biology. Moreover, we believe that your study will also reinforce and expand our understanding of Wnt signaling in kidney development.

**Decision letter after peer review:**

Thank you for submitting your article "A β-catenin-driven switch in TCF/LEF transcription factors promotes commitment of mammalian nephron progenitor cells" for consideration by *eLife*. Your article has been reviewed by three peer reviewers, and the evaluation has been overseen by Edward Morrisey as the Senior and Reviewing Editor. The following individual involved in review of your submission has agreed to reveal their identity: Melissa H Little (Reviewer #1).

The reviewers have discussed the reviews with one another and the Editor has drafted this decision to help you prepare a revised submission.

As the editors have judged that your manuscript is of interest, but as described below that additional experiments may be required before it is published, we would like to draw your attention to changes in our revision policy that we have made in response to COVID-19 (https://elifesciences.org/articles/57162). First, because many researchers have temporarily lost access to the labs, we will give authors as much time as they need to submit revised manuscripts. We are also offering, if you choose, to post the manuscript to bioRxiv (if it is not already there) along with this decision letter and a formal designation that the manuscript is "in revision at *eLife*". Please let us know if you would like to pursue this option. (If your work is more suitable for medRxiv, you will need to post the preprint yourself, as the mechanisms for us to do so are still in development.)

Most of the requested revisions can be addressed without the need for additional experiments. Of note, the editors do not feel that the request for additional Western blotting noted by reviewer 2 is required for the revision.

Reviewer #1:

This is a detailed and careful analysis of the mechanism by which cultured mouse nephron progenitors respond to low versus high levels of CHIR, a GSK3β inhibitor used to mimic canonical Wnt activation. It includes transcriptional profiling, ATAC-Sed ChIP seq and Hi-C analyses. The work is a collaboration between two groups who have established the duality of the action of Wnt signalling on nephron progenitors at the cellular and molecular level and developed protocols for their isolation and maintenance in vitro using a support medium called NPEM. Having titrated two concentrations of CHIR as representing the low (NPC maintenance ) and high (commitment) inducing states, they investigate how this translates into transcriptional output in vitro, revealing a shift in the interactions between b-catenin and specific TF complexes at a range of target genes (associated with either self-renewal or commitments) and novel genes. They relate this back in some instances to previously proposed targets. The work is logical, careful, comprehensive and extensive. The claims made are well supported by the data. In essence, this study both supports the authors prior findings but clarifies the interaction of b-catenin with specific chromatin complexes and target genes. There is no particular need to ask for any further studies.

Reviewer #2:

I am not a bioinformation and I will therefore restrict my comments to the biology behind the data. In general the manuscript is very well written and the majority of conclusions drawn supported by the data. However, there are a number of points that need to be clarified.

1) The study is based on MACS isolated cells that are treated with varying concentrations of CHIR, but I was unable to find a detailed description of the experimental set up. How long were the cells cultured before they were harvested for the analysis? What was the plating density? These are important parameters that can dramatically influence the cellular response. Please provide a dedicated paragraph in the Materials and methods section.

2) Figure 1A evaluates expression of different proteins by inferring levels from immunofluorescence intensity. Western blots are much more accurate for a quantitative analysis and should be shown instead. (A Western blot for β-catenin is actually shown in Supplementary Figure 3C, but no quantification is provided).

3) Axin2 is considered a direct target of β-catenin and is often used as a read out for canonical signaling. Axin2 appears to be expressed in renal progenitors, albeit at relatively low levels (Vidal et al., 2020). How does expression, occupation of enhancer sites and looping at this gene change under the different conditions?

4) The Hi-C studies allows drawing interesting conclusions about chromatin looping under two different CHIR conditions and clearly shows that certain genes are "primed" on the chromatin level to engage in transcription, whereas others only make contact with their enhancers upon High CHIR treatment. In this context, Figure 5C examines the distal enhancer of Wnt4 and the authors demonstrate a clear switch of enhancer occupation from TCF7l1/2 and SIX2 towards TCF7 and LEF1. β-catenin seems already bound to this enhancer under low CHIR conditions, but increases under High CHIR (this is also stated in the subsection “β-catenin uses both pre-established and de novo enhancer-promoter loops to drive NPC differentiation program”). Figure 5D, however, suggests that under low CHIR conditions b-catenin does not bind to the WNT-DE. This is confusing and needs to be clarified

5) Schuijers et al., 2014, identified two classes of β-catenin binding sites in murine intestine and two different cell lines and concluded that the vast majority of β-catenin transcriptional activity is mediated through TCF/LEF transcription factors. This paper needs to be cited and discussed.

6) The study uses CHIR to mimic different levels of β-catenin activity. CHIR is an inhibitor of GSK3β, a central component of a protein destruction complex that not only influences β-catenin stability, but also other signaling and cellular pathways. The authors acknowledge this shortcoming in their Discussion and suggest themselves that "Wnt-ligand mimetics offer a promising future approach to confirm CHIR-β-catenin-centered findings". The paper would gain impact, if some of the most important findings could be verified by testing target site occupation on some key sites using WNT surrogates (ChIP-qPCR experiments).

7) Under low CHIR conditions only 234 β-catenin ChIP-seq peaks were detected and the authors hypothesize that WNT signaling may regulate proliferation (at least partly) independently of DNA binding. In this context the Wnt/STOP model should be discussed (DOI: 10.1016/j.molcel.2014.04.014), a mechanism that protects proteins from GSK3-dependent degradation, but does not seem to involve β-catenin activity.

Reviewer #3:

Aspects of the manuscript that require further clarification/revision:

1) Details on in vitro NPC culture conditions and incubation hours with low or high CHIR are not provided and could be included in the Materials and methods. This information is important when different studies are compared.

2) Data presented in supplementary files

Supplementary files should contain more details, including titles with a description of the data shown, abbreviations, as the authors have shown under file "Notes" in Supplementary file 3 or 6

– Supplementary file 2 is not cited in the text (likely in the first Chapter of Results?).

– Concerning Supplementary file 3, cited several times in the manuscript, it is hard to find the results indicated.

To simplify inspection of the lists of DE genes shown, it would be helpful to add an additional file or supplementary file with selected differentially expressed genes for each condition indicating the fold-change cut off value applied ( >1 or other) and the adjusted p-value (FDR).

3) Related to the previous comment, Figure 1D and Figure 1—figure supplement 1 show the Top 5 enriched GO terms on the basis of lists of selected transcripts that are not shown (i.e. NPC>NFC= 241 transcripts, High CHIR > Low CHIR = 194 transcripts, Low CHIR> High CHIR =208 transcripts…).

Similarly Figure 1—figure supplement 1 shows GO terms of Low CHIR>NPC and NPC >Low CHIR of additional lists of transcripts. In addition is indicated "* genes were selected with a different DE threshold" but it is not shown which is such threshold (Panel E).

4) It is unclear how the selection of the Top5 GO-terms was performed because in many cases they are redundant Figure 1—figure supplement 1D Low CHIR > NPC (494 transcripts): top 5 GO terms are redundant as they are in Figure 4—figure supplement 2E.

5) The authors indicate that "high CHIR led to a down regulation of regulators and markers of self-renewing NPCs, including Six2 (Self et al., 2006), Cited1 (Mugford et al., 2009), Osr1 (Xu, Liu, et al. 2014) and Eya1 (Xu, Wong, et al. 2014), and a concomitant increase in expression of genes associated with induction of nephrogenesis, such as Wnt4, Jag1, Lhx1, Pax8 and Fgf8 (Figure 1C; Park et al., 2007). Trends in gene expression from mRNA-seq were confirmed by RT-qPCR analysis (Figure 1—figure supplement 1B)". However this is not fully supported by the data presented in Figure 1C, Supplementary file 3 and Figure 1—figure supplement 1B, which do not show a significant increase in Pax8 and Ffg8 transcript levels.

Please comment whether these differences in the expression of these differentiated genes in High CHIR are due either to the different ex vivo inductive culture conditions or another alternative explanation.

6) “Figure 2—figure supplement 1. ATAC-Seq data analysis –(A) Hierarchical cluster of R-square values between normalized ATAC-Seq reads within merged peaks from all samples.”

To obtain a more global picture of the process, it would further informative if the genome-wide ATAC-sequencing heatmaps are associated to the corresponding GO analysis on an additional column highlighting the signal pathways of each condition.

Indeed, NPC self-renewing is known to depend on additional growth factor signaling pathways, such as FGF and BMP. The interactions of the Wnt-pathway with other spatially and temporally restricted signaling pathways may in addition affect the effects of Wnt-β-catenin on the expansion of nephron progenitors, which the authors found here to be independent of direct β-catenin/chromatin engagement and probably involving other mechanisms.

7) Supplementary Figure 3C. Immunoblots of TCF/LEF family factors in NPC cultures should be improved (in particular for Tcf7, Tcf7l1 detection over background), and the MW size markers indicated

8) A recent publication of Hilliard et al., 2019 Development reported the differential chromatin accessibility in mouse NPCs at embryonic day E13/ E16 and postnatal P2 and as well as the chromatin landscape in young and old NPCs. This study is not commented. The NPC culture conditions used by Hilliard et al. for ChIP-seq analyses were different than those used here (and in addition not well detailed), limiting the comparisons. However, ATAC-seq was performed on freshly isolated E16 NPCs. It would be important to discuss and compare these previous results of Hilliard et al. with those obtained here.

---

## [Author Response]

Reviewer #2:I am not a bioinformation and I will therefore restrict my comments to the biology behind the data. In general the manuscript is very well written and the majority of conclusions drawn supported by the data. However, there are a number of points that need to be clarified.1) The study is based on MACS isolated cells that are treated with varying concentrations of CHIR, but I was unable to find a detailed description of the experimental set up. How long were the cells cultured before they were harvested for the analysis? What was the plating density? These are important parameters that can dramatically influence the cellular response. Please provide a dedicated paragraph in the Materials and methods section.

Indeed, these are critical points, we apologize for not having this information in the original manuscript and thank the reviewer for spotting this omission. The relevant information is now included in the Materials and methods section of the revised manuscript.

2) Figure 1A evaluates expression of different proteins by inferring levels from immunofluorescence intensity. Western blots are much more accurate for a quantitative analysis and should be shown instead. (A Western blot for β-catenin is actually shown in Supplementary Figure 3C, but no quantification is provided).

While Western blots give a population-based view, each cell is its own data point, and quantifying single is a reasonable approach as evidenced by the wealth of quantitative approaches at the single cell level. The Western blot in Supplementary Figure 3C has now been quantified and placed in the new Figure 3—figure supplement 2. The result is consistent with the original conclusion from immunostaining data in the original manuscript (Figure 3C).

3) Axin2 is considered a direct target of β-catenin and is often used as a read out for canonical signaling. Axin2 appears to be expressed in renal progenitors, albeit at relatively low levels (Vidal et al., 2020). How does expression, occupation of enhancer sites and looping at this gene change under the different conditions?

Referring to new Supplementary file 2 (replacement of old Supplementary file 3), the mean TPM for Axin2 transcript is 4.6 in low CHIR and 41.7 in high CHIR: an approximately 10-fold change with an adjusted p-value of 1e-17. In line with this trend, TPM of Axin2 is 1.0 in no CHIR condition, which is significantly lower than that in the low CHIR condition (p = 1e-24). The data is consistent with an activation of canonical Wnt signaling activation. We did observe strong binding of β-catenin, Tcf7 and Lef1 near Axin2 promoter and a distal site 30 kb upstream in high CHIR condition only. This agrees with RNA PolII loading and H3K27 acetylation of the promoter region in high but not low CHIR. Tcf7l2 binding is present in both conditions but weaker in the high CHIR condition. However, we did not capture a loop to the Axin2 promoter in either condition in the Hi-C date. A caveat of Hi-C studies is an under-representation of promoter-proximal interactions which are difficult to distinguish from the high expectation of interactions for near-adjacent genomic sequences. For readers’ convenience, we provided a table of all loops connecting to at least one gene’s promoter (new Supplementary file 5).

4) The Hi-C studies allows drawing interesting conclusions about chromatin looping under two different CHIR conditions and clearly shows that certain genes are "primed" on the chromatin level to engage in transcription, whereas others only make contact with their enhancers upon High CHIR treatment. In this context, Figure 5C examines the distal enhancer of Wnt4 and the authors demonstrate a clear switch of enhancer occupation from TCF7l1/2 and SIX2 towards TCF7 and LEF1. β-catenin seems already bound to this enhancer under low CHIR conditions, but increases under High CHIR (this is also stated in the subsection “β-catenin uses both pre-established and de novo enhancer-promoter loops to drive NPC differentiation program”). Figure 5D, however, suggests that under low CHIR conditions b-catenin does not bind to the WNT-DE. This is confusing and needs to be clarified

We thank the reviewer for pointing out this discrepancy. We have adjusted Figure 5D to make it reflect the data where weak β-catenin binding is present in some sites in low CHIR condition.

5) Schuijers et al., 2014, identified two classes of β-catenin binding sites in murine intestine and two different cell lines and concluded that the vast majority of β-catenin transcriptional activity is mediated through TCF/LEF transcription factors. This paper needs to be cited and discussed.

We thank the reviewer for bringing this point to our attention. We have modified the text in describing the NPC ChIP-seq data in the Results as follows: “In both low and high CHIR conditions, β-catenin association overlapped extensively with the binding of cognate TCF/LEF factors specifically enriched in each condition (Figure 4-figure supplement 1B). […] This is consistent with previous findings in murine intestinal studies that localization of β-catenin is primarily dependent on TCF/LEF factors (Schuijers et al., 2014)”.

6) The study uses CHIR to mimic different levels of β-catenin activity. CHIR is an inhibitor of GSK3β, a central component of a protein destruction complex that not only influences β-catenin stability, but also other signaling and cellular pathways. The authors acknowledge this shortcoming in their Discussion and suggest themselves that "Wnt-ligand mimetics offer a promising future approach to confirm CHIR-β-catenin-centered findings". The paper would gain impact, if some of the most important findings could be verified by testing target site occupation on some key sites using WNT surrogates (ChIP-qPCR experiments).

A major strength of the current work is an extensive analysis of large, unbiased datasets. It is not clear what value is really added by a limited and selected analysis of Wnt-ligand mimetics to the current work? A thorough evaluation of small-molecule versus protein mimetic is better suited to an independent future study.

7) Under low CHIR conditions only 234 β-catenin ChIP-seq peaks were detected and the authors hypothesize that WNT signaling may regulate proliferation (at least partly) independently of DNA binding. In this context the Wnt/STOP model should be discussed (DOI: 10.1016/j.molcel.2014.04.014), a mechanism that protects proteins from GSK3-dependent degradation, but does not seem to involve β-catenin activity.

We appreciate the reviewer bringing this interesting reference to our attention. Indeed, the work provides evidence of CHIR action that is independent of β-catenin and might contribute to the pro-proliferation function of Wnt signaling. We have referenced this paper in the Discussion: “Additionally, CHIR-mediated inhibition of GSK3 outside of β-catenin regulation might play a role …”.

Reviewer #3:Aspects of the manuscript that require further clarification/revision:1) Details on in vitro NPC culture conditions and incubation hours with low or high CHIR are not provided and could be included in Materials and methods. This information is important when different studies are compared.

See response to reviewer 2 – we added this information to the Materials and methods and thank the reviewer for pointing out the omission.

2) Data presented in supplementary filesSupplementary files should contain more details, including titles with a description of the data shown, abbreviations, as the authors have shown under file "Notes" in Supplementary file 3 or 6– Supplementary file 2 is not cited in the text (likely in the first Chapter of Results?).– Concerning Supplementary file 3, cited several times in the manuscript, it is hard to find the results indicated.To simplify inspection of the lists of DE genes shown, it would be helpful to add an additional file or supplementary file with selected differentially expressed genes for each condition indicating the fold-change cut off value applied ( >1 or other) and the adjusted p-value (FDR).

Supplementary file 2 is a TPM table that lists TPM values of all genes observed in each sample. The data from this table was used in 1C, 3A and S3B, and can be used independently of any specifically mentioned data in this paper for interested readers. To make it clearer and more reader friendly, in this revision we merged Supplementary file 2 and 3 to generate a new Supplementary file 2, and we have added more description of the table in the “Notes” sheet which are intended to make the data clearer to the reader. Any reader can search for name of genes they are interested in and see their expression levels, fold changes and p-values between two specific conditions, referring to the “Notes” sheet.

3) Related to the previous comment, Figure 1D and Figure 1—figure supplement 1 show the Top 5 enriched GO terms on the basis of lists of selected transcripts that are not shown (i.e. NPC>NFC= 241 transcripts, High CHIR > Low CHIR= 194 transcripts, Low CHIR> High CHIR=208 transcripts…).Similarly Figure 1—figure supplement 1 shows GO terms of Low CHIR>NPC and NPC >Low CHIR of additional lists of transcripts. In addition is indicated "* genes were selected with a different DE threshold" but it is not shown which is such threshold (Panel E).

We thank the reviewer for pointing out these omissions. The threshold used in Figure 1—figure supplement 1E has been added to the new Supplementary file 2 “Notes” sheet. We identified some discrepancies between in the GO term figures (Figure 1D and Figure 1—figure supplement 1D and E) and Supplementary file 2 and have now corrected these and added adjusted p-values instead of p-values in the figures mentioned. There are no changes in the conclusions from correcting these errors.

4) It is unclear how the selection of the Top5 GO-terms was performed because in many cases they are redundant Figure 1—figure supplement 1D Low CHIR > NPC (494 transcripts): top 5 GO terms are redundant as they are in Figure 4—figure supplement 2E.

We are not sure what the reviewer refers to as redundant, as we did not see any redundancy among the GO terms. Also, Figure 4—figure supplement 2E does not contain any GO terms; maybe there is a typo here. The selection to Top5 GO terms is simply by the rank of their p-value or adjusted p-values from low to high.

5) The authors indicate that " high CHIR led to a down regulation of regulators and markers of self-renewing NPCs, including Six2 (Self et al., 2006), Cited1 (Mugford et al., 2009), Osr1 (Xu, Liu, et al., 2014) and Eya1 (Xu, Wong, et al., 2014), and a concomitant increase in expression of genes associated with induction of nephrogenesis, such as Wnt4, Jag1, Lhx1, Pax8 and Fgf8 (Figure 1C; Park et al., 2007). Trends in gene expression from mRNA-seq were confirmed by RT-qPCR analysis (Figure 1—figure supplement 1B)". However this is not fully supported by the data presented in Figure 1C, Supplementary file 3 and Figure 1—figure supplement 1B, which do not show a significant increase in Pax8 and Ffg8 transcript levels.Please comment whether these differences in the expression of these differentiated genes in High CHIR are due either to the different ex vivo inductive culture conditions or another alternative explanation.

Fgf8 is significantly upregulated in the high CHIR condition as compared to in the low CHIR condition (p-adj = 0.0001; Figure 1C, Figure 1—figure supplement 1B and new Supplementary file 2). However, it was not identified as a “differentially expressed gene” in this comparison because the TPM of the highest transcript did not pass 5 – our criteria for inclusion (explained in the Materials and methods and new Supplementary file 2) though the change in expression is significant. For Pax8, we agree, this is an error, there is no significant change between low and high CHIR conditions and we have accordingly corrected the statement. In contrast, there is a significant change in Pax8 expression between the low CHIR and no CHIR conditions (p-adj <0.001; new Supplementary file 2), which suggests Pax8 expression maybe hyper-sensitive to CHIR concentration. Interestingly, mouse genetic studies suggest up-regulation of Pax8 is the first response in NPC commitment.

6) “Figure 2—figure supplement 1. ATAC-Seq data analysis – (A) Hierarchical cluster of R-square values between normalized ATAC-Seq reads within merged peaks from all samples.”To obtain a more global picture of the process, it would further informative if the genome-wide ATAC-sequencing heatmaps are associated to the corresponding GO analysis on an additional column highlighting the signal pathways of each condition.Indeed, NPC self-renewing is known to depend on additional growth factor signaling pathways, such as FGF and BMP. The interactions of the Wnt-pathway with other spatially and temporally restricted signaling pathways may in addition affect the effects of Wnt-β-catenin on the expansion of nephron progenitors, which the authors found here to be independent of direct β-catenin/chromatin engagement and probably involving other mechanisms.

We understand the reviewer’s intention to bring together the GO terms and a genomic manifestation of the ATAC-Seq data, which might enhance the visual perception. However, the heatmap (Figure 2—figure supplement 1A) shows the R-squared values between each sample and does not reflect information on particular regions. On the other hand, the GO term results shown in Figure 2C and Figure 2—figure supplement 1C were based on the differentially accessible regions between the specified conditions. Therefore, there is not a direct connection between the heatmap and the GO terms. Instead, we came up with heatmaps showing signal of the DA regions across sample and attached corresponding GO terms and motifs found and added this to a revised Figure 2C with the intention of generating a visual representation similar to the reviewer’s suggestion. We have also improved our workflow and Supplementary file 3 for better statistical rigor and making the data more accessible to the reader with accompanying minor changes in Figure 2 and figure 2—figure supplement 1.

7) Supplementary Figure 3C. Immunoblots of TCF/LEF family factors in NPC cultures should be improved (in particular for Tcf7, Tcf7l1 detection over background), and the MW size markers indicated

We have included the MW size marker in the immunoblots (new Figure 3—figure supplement 2). Although the Tcf7 result looks reasonable to us, we are aware there are background bands for Tcf7l1. We have tried several different antibodies but this one gives the best result. Unfortunately, there isn’t anything we can think of to improve the result.

8) A recent publication of Hilliard et al., 2019 Development reported the differential chromatin accessibility in mouse NPCs at embryonic day E13/ E16 and postnatal P2 and as well as the chromatin landscape in young and old NPCs. This study is not commented. The NPC culture conditions used by Hilliard et al. for ChIP-seq analyses were different than those used here (and in addition not well detailed), limiting the comparisons. However, ATAC-seq was performed on freshly isolated E16 NPCs. It would be important to discuss and compare these previous results of Hilliard et al. with those obtained here.

We now refer to the Hilliard et al., 2019 study in the revised manuscript in “CHIR-mediated induction modifies the epigenomic profile of NPCs” part of the Results section: “Statistical assessment (Supplementary file 2) determined…”. In the Hilliard et al., 2019 study, a major focus was the identification of epigenetic differences that may relate to maturation of the Six2+ NPC state between E16 and P2, shortly before the NPC population is lost. Consequently, ATAC-seq data in this work aimed to identify epigenetic differences reflecting elevated Wnt/ β-catenin signaling activity as NPCs underwent developmental aging. While there are some expected similarities, i.e., an enhanced differentiation signature in P2 vs. E16 also observed in high CHIR vs. low CHIR condition. Importantly, the in vivo data are interpreted as reflecting, at least in part, elevated Wnt-signaling. However, this is an interpretation in contrast to our study which directly modulates the canonical Wnt-signaling pathway to determine an effect on NPCs. Given these differences, we have not incorporated additional analyses of the Hilliard et al., 2019 data.